# Understanding Robustness Against Gradient Inversion Attacks: A Flat Minima Perspective

## Abstract

Gradient Inversion Attacks (GIAs), which aim to reconstruct input data from its gradients, pose substantial risks of data leakage and challenges of data privacy in distributed learning systems, e.g., federated learning (FL). Nevertheless, existing defenses against GIA are mostly ad-hoc by relying on gradient modifications without a principle of when gradients are vulnerable to GIA and how we can fundamentally suppress the possibility of data leakage. We interpret GIA with the mutual information between the gradients $G$ and their data $X$, i.e., $I(X;G)$, which is revealed to be upper-bounded by the Hessian of loss. Based on the findings, we rethink the robustness against GIA for a flat minima searching-based FL algorithm, where it inherently suppresses Hessian values, thus minimizing $I(X;G)$. We extensively demonstrate that the gradients computed by searching flatter minima in the FL scenario achieve a substantial improvement in robustness against GIAs. Our work sheds light on novel benefits of flat minima searching, not only promoting better generalization but also hardening privacy in FL systems.

## 1 Introduction

Federated Learning (FL) has emerged as one of the most promising frameworks for decentralized training. The primary objective of FL is to enable model training on individual client devices without transmitting data to a central server, thereby mitigating privacy risks such as data leakage. In this paradigm, each client trains a local model, and the server aggregates these locally computed model updates to construct a global model (McMahan et al., 2017). Nevertheless, recent studies have revealed that FL remains vulnerable to significant privacy risks. In particular, gradients from local models can leak sufficient information about client data via an attacking mechanism widely known as Gradient Inversion Attacks (GIAs) (Zhu et al., 2019; Geiping et al., 2020; Li et al., 2022).

As baselines for mitigating these GIA threats, prior approaches directly distort the gradient values, e.g., adding random Gaussian noise to gradients (Geyer et al., 2017) or noising the original data itself (Sun et al., 2021), thereby leading to insufficient reconstruction by GIA. Alternatively, gradient sparsification by partially dropping (Aji & Heafield, 2017) or clipping (Wei et al., 2021) the gradient values has been shown to impede reconstruction. However, these approaches primarily modify the gradient or the data themselves, thereby not only hindering reconstruction but also taking a risk of wrongly altering the training. More importantly, prior approaches lack of principled way of knowing when gradients become vulnerable to GIA and suppressing the possibility of data leakage via GIA. These limitations leave the existing defensive ways still vulnerable to the novel advanced attacks (Mo et al., 2021; Liu et al., 2021).

To establish a fundamental understanding, we frame GIA through a lens of mutual information between the gradients $G$ and the corresponding data $X$, i.e., $I(X;G)$. Because gradients inherently encode the direction and magnitude of parameter updates toward an optimal point in the loss landscape, $G$ naturally contains substantial information about the underlying data $X$, formalized by the high mutual information with the data. We conjecture that the ground truth of gradients is indeed an ensemble across datasets; thus, it does not have to form a one-to-one mapping to each sample. Therefore, we argue that reducing the sensitivity of $G$ to $X$ instances, equivalently lowering $I(X;G)$, is fundamentally feasible, making it more difficult for adversaries to reconstruct $X$ from $G$ while keeping the distributed training undamaged.

In this work, we formulate an upper bound of $I(X;G)$ in terms of the empirical Hessian, thereby revealing the connection between $I(X;G)$ and the curvature of the loss landscape. By pursuing a flatter loss surface during training, i.e., keeping the Hessian minimal, we can suppress $I(X;G)$ to defend against GIAs. Based on this intuition, we rethink FL with flat minima searching, such as FedSAM (Qu et al., 2022a), FedASAM (Caldarola et al., 2022), FedGF (Lee & Yoon, 2024), etc., with a view to tightly bounding $I(X;G)$ and ultimately hindering GIA. Finally, we rigorously prove that convergence toward flat minima in FL ensures robustness against GIA while achieving stable convergence. Our contributions are summarized as follows:

- **A mutual information perspective of GIA:** We establish an upper bound on the mutual information $I(X;G)$ by explicitly associating it to the empirical Hessian $\boldsymbol{H}$. Our key lessons are: **i)** A larger Hessian makes gradients vulnerable to GIAs, and **ii)** Gradients computed on a smooth loss surface can be a remedy for GIAs.

- **Rethinking of flat minima for defending GIA:** We argue that flat minima searching paves a smooth path toward minima, thereby resulting in suppressed $I(X;G)$ and hindering GIA.

- **Convergence and robustness guarantees:** We rephrase the convergence analysis of Fed-SAM, which is a simple baseline of flat minima searching FL, so that it theoretically guarantees the robustness against GIA during the training phase.

- **Benchmarking the existing defenses and attacks:** By an extensive investigation of the existing defensive FL and GIAs, we demonstrate that the robustness correlates with Hessian, and flat minima suppress the Hessian spectrum, reducing attack performance.

## 2 RELATED WORK

We introduce prior studies on federated learning (FL), gradient inversion attacks (GIAs), defense mechanisms, and flat minima-based optimization methods to establish the context of this work.

**Federated Learning:** FL enables multiple clients to collaboratively train a global model distributed from a central server without revealing their local datasets (McMahan et al., 2017). Each client computes updates on its local datasets and transmits resulting updates (e.g., parameters or gradients) to the central server, which aggregates and updates to refine the global model. This framework mitigates the risk of direct data leakage from the central server. Nevertheless, recent studies have demonstrated that gradients during communication still encode sensitive information about local data, thereby allowing adversaries to reconstruct training data.

**Gradient Inversion Attacks:** We investigate an honest-but-curious server in GIA. In this setting, the adversary exploits shared gradients to reconstruct private training data. Early studies propose optimization-based approaches, such as Deep Leakage from Gradients (DLG) (Zhu et al., 2019), which iteratively optimize dummy inputs to match observed gradients. iDLG (Zhao et al., 2020) extends DLG by identifying ground-truth labels from last-layer gradients, enabling more accurate reconstruction. Geiping et al. (2020) adopts cosine similarity for high-fidelity recovery of input data. While these optimization-based methods rely on gradient matching, subsequent studies further propose generative-model-based methods that leverage priors from generative model for reconstruction.

The generative-model-based approaches utilize priors to improve data reconstruction. GIAS (Jeon et al., 2021) optimizes latent representation of models with observed gradients, while GGL (Li et al., 2022) employs Generative Adversarial Networks (GAN) priors to reconstruct private data under defense mechanisms. GIFD (Fang et al., 2023) extends to out-of-distribution settings by optimizing intermediate feature layers, undermining the assumption that GANs and FL operate on the same data distribution. These studies indicate that GIA evolves into highly effective threats, diminishing the effectiveness of basic defense strategies.

**Defense Strategies Against GIA:** To counteract GIA, several defense strategies have been proposed. Differential privacy (Geyer et al., 2017) perturbs gradients with stochastic noise, while Soteria (Sun et al., 2021) adds noise to data representations. Other approaches, such as gradient sparsification (Aji & Heafield, 2017) and clipping (Wei et al., 2021), selectively truncate or mask gradient components to limit information leakage. Although these methods hinder gradient-based reconstruction, they remain vulnerable to generative-model-based attacks. More importantly, most strategies directly distort gradients or data while overlooking the intrinsic sensitivity of gradients to

training data, the fundamental source of information leakage. In this work, we address this issue by establishing a theoretical connection between gradient sensitivity and mutual information.

**Flat Minima and Privacy in Federated Learning:** Flat minima searching (Hochreiter & Schmidhuber, 1997), which characterizes the tendency of models to converge toward regions where the loss remains stable under small perturbation, has been widely recognized as a key factor for model generalization. Sharpness-Aware Minimization (SAM) (Foret et al., 2020) formalizes this idea by explicitly seeking flat minima through min-max optimization of local perturbations within a neighborhood of the parameters. Extension of SAM to FL setting, including FedSAM (Qu et al., 2022a), FedASAM (Caldarola et al., 2022), and FedGF (Lee & Yoon, 2024) demonstrate improved optimization stability under heterogeneous data distribution and enhance generalization by analyzing optimization trajectories and gradient variance (Jastrzebski et al., 2020). Beyond generalization, several studies integrate flat minima with differential privacy (DP) to mitigate the negative impact of noise on training while preserving optimization ability (Park et al., 2023; Wang et al., 2025). However, a principled theoretical understanding of how flatness relates to gradient leakage, particularly in FL, remains unexplored. In this work, we establish a formal connection between flat minima and mutual information, showing that flat minima inherently strengthen model robustness against GIA.

## 3 ROBUSTNESS AGAINST GRADIENT INVERSION ATTACK VIA FLAT MINIMA

In this section, we provide a fundamental understanding of how the mutual information between gradients and input has been bounded by the Hessian of the loss, thus leading to a theoretical analysis that demonstrates how FedSAM prevents GIA attacks.

**Notations:** $\mathbb{R}$ means the real number set. The training data and labels are given by $X \in \mathbb{R}^{m \times n}$ and $Y \in \mathbb{R}^m$. A batch is represented as $(X_B, Y_B) = (x_i, y_i)_{i=1}^B$, where $B$ is the batch size and $\hat{X} \in \mathbb{R}^{m \times n}$ states the reconstructed data from an adversary. The per-sample loss is defined as $\mathcal{L}(\theta; x, y) = -\log p(y \mid x; \theta)$, i.e., the cross-entropy loss. The batch loss is $\mathcal{L}_B(\theta; X_B, Y_B) = \frac{1}{B} \sum_{i=1}^B \mathcal{L}(\theta; x_i, y_i)$. Let $\theta$ indicate the local parameter, whose model is $f_\theta : X \mapsto G$, and the attack model is $g_\phi : G \mapsto \hat{X}$ parameterized by $\phi$. The observed gradient is $G = \nabla_\theta \mathcal{L}_B(\theta; X_B, Y_B) \in \mathbb{R}^d$ and the Hessian is $\boldsymbol{H}_B = \nabla_\theta^2 \mathcal{L}_B(\theta; X_B, Y_B) \in \mathbb{R}^{d \times d}$. For per-sample gradient $g_i = \nabla_\theta \mathcal{L}(\theta; x_i, y_i)$, the covariance of the batch gradient is $\Sigma_G = \frac{1}{B} \sum_{i=1}^B (g_i - G)(g_i - G)^\top$.

**FL Problem Settings:** We consider FL setting with $N$ clients. Each client $i \in [N]$ possesses private data distribution $D_i$, which may differ across clients due to data heterogeneity. A sample from client $i$ is denoted as $\xi_{ij} = (x_{ij}, y_{ij}) \sim D_i$. At each communication round $r \in [R]$, the server randomly selects a subset $S^r \subseteq [N]$ of clients and broadcasts the current global model $\theta^r$ to them. Each selected client performs $K$ steps of local updates with a mini-batch of size $B$ and learning rate $\eta_l$. Specifically, the local empirical risk for client $i$ is defined as $F_i(\theta) := \mathbb{E}_{\xi \sim D_i} [\mathcal{L}(\theta; \xi)]$. After local training, clients send their model updates $\Delta_i^r$ to the server. The server then aggregates the updates and obtains the new global model $\theta^{r+1}$ with global learning rate $\eta_g$: $\theta^{r+1} = \theta^r + \eta_g \Delta^r$, $\Delta^r = \frac{1}{s} \sum_{i \in S^r} \Delta_i^r$. Consequently, FL minimizes the global objective $F(\theta)$:

$$\theta^* = \arg\min_\theta \{F(\theta) := \sum_{i \in [N]} \frac{m_i}{m} F_i(\theta)\}, \tag{1}$$

where $m_i$ is the number of samples in client $i$, and $m = \sum_{i \in [N]} m_i$. In the case of FedSAM, each local update is modified by perturbing the parameters with radius $\rho$, based on the SAM principle.

### 3.1 BOUNDING MUTUAL INFORMATION VIA EMPIRICAL HESSIAN

We analyze mutual information $I(X; G)$, which quantifies how much information the gradients $G$ encode about the training data $X$. This information-theoretic perspective has been employed in prior work to evaluate privacy leakage (Liu et al., 2021; Mo et al., 2021). From this standpoint, we interpret $I(X; G)$ through the entropy of the gradients $H(G)$, since the amount of information preserved in $G$ is reflected in its uncertainty. However, the exact computation of $H(G)$ is intractable, we consider the worst-case uncertainty instead, which is bounded above by the entropy of the multivariate

Gaussian distribution with covariance matrix $\Sigma_G$:

$$H(G) \leq \frac{d}{2} \log(2\pi e) + \frac{1}{2} \log \det(\Sigma_G), \tag{2}$$

where $\Sigma_G \succ 0$ ensures that $\log \det(\Sigma_G)$ is well defined.

**Theorem 3.1 (Bound on Mutual Information with Empirical Hessian and Batch Size)**
*Consider the score function $s(\theta; x, y) = \nabla_\theta \log p(y \mid x; \theta)$, which is the negative gradient of loss function. The covariance of gradients $\Sigma_G$ can be related to the empirical Hessian, leading to following upper bound on the mutual information $I(X; G)$.*

$$I(X; G) \leq \frac{d}{2} \log(\frac{2\pi e}{B}) + \frac{1}{2} \log\left(\det(\boldsymbol{H})\right), \tag{3}$$

Theorem 3.1 shows that the determinant of the empirical Hessian fundamentally governs mutual information. A smaller Hessian, which corresponds to flatter curvature, reduces the information encoded in the gradients and strengthens robustness to the gradient inversion attack, hindering data reconstruction. In addition, the batch size $B$ serves as a mitigating factor, increasing $B$ tightens the mutual information bound and enhances robustness. The proof is presented in Appendix B.1.

### 3.2 ROBUSTNESS THROUGH SHARPNESS-AWARE MINIMIZATION

We then step forward to elucidate how Sharpness-Aware Minimization (SAM) (Foret et al., 2020), a principled method for finding flatter minima, smoothly renders the loss surface curvature captured by the Hessian, leading to a tightened mutual information bound. As a preliminary, the objective function of SAM is formalized as follows:

$$F_{\text{SAM}}(\theta) := \sum_i^N \frac{m_i}{m} F_i^{\text{SAM}}(\theta), \qquad F_i^{\text{SAM}}(\theta) := \max_{||\delta||_2 \leq \rho} F_i(\theta + \delta).$$

**Theorem 3.2 (Mutual Information Bound via Sharpness-Aware Minimization)** *Considering $\delta = \rho v$, $\|v\|_2 = 1$ and a second-order Taylor expansion of the SAM objective, we can derive the following upper bound of $I(X; G)$:*

$$I(X; G) \leq \frac{d}{2} \log\left(\frac{2\pi e}{B}\right) + \frac{1}{2} \log \det(\boldsymbol{H}) \tag{4}$$

$$\leq \frac{d}{2} \log\left(\frac{2\pi e}{B}\right) + \frac{d}{2} \log\left(\frac{2}{\rho^2}\left(F_{SAM}(\theta) - F(\theta) + \rho\|G\|_2\right)\right) \tag{5}$$

*where $B$ is the batch size, $d$ is the number of model parameters, $F_{SAM}$ the SAM objective function.*

In Theorem 3.2, the SAM objective $F_{\text{SAM}}$ appears explicitly in the bound. When the training objective $F$ becomes $F_{\text{SAM}}$, i.e., adopting flatter minima searching, it directly reduces the empirical Hessian and $I(X; G)$. Moreover, the batch size $B$ tightens the bound. From a robustness perspective, this indicates that gradient computed under FedSAM carry less specific information about input data, mitigating gradient leakage. In other words, learning toward flatness enhances model generalization and strengthens robustness against gradient inversion attacks.

**Convergence Analysis of FL with Batch size and Hessian Perspectives:** The theoretical bounds in Theorem 3.1 and Theorem 3.2 reveal that the determinant of the empirical Hessian $\det(\boldsymbol{H})$ and the batch size $B$ are the key quantities to mitigate GIA. We subsidiarily verify these quantities in the optimization dynamics, and among the family of SAM-based methods, we focus on FedSAM, which directly reflects our targeting FL system. We reformulate the convergence analysis of FedSAM done in Qu et al. (2022a) by using the term of the empirical Hessian.

For the case of full client participation with learning rates $\eta_l = \mathcal{O}(\frac{1}{\sqrt{R}KL}), \eta_g = \sqrt{KN}$ and the perturbation $\rho = \mathcal{O}(\frac{1}{\sqrt{R}})$, the iterates generated by FedSAM satisfy:

$$\mathcal{O}\left(\frac{FL}{\sqrt{RKN}} + \frac{\sigma_g^2}{R} + \frac{L^2\sigma_l^2}{R^{3/2}\sqrt{KN}} + \frac{L}{BK^{3/2}\sqrt{RN}} + \frac{\det(\boldsymbol{H})}{BK^{3/2}L'\sqrt{RN}}\right)$$

For partial client participation with $S \geq K$, and learning rates $\eta_l = \mathcal{O}(\frac{1}{\sqrt{R}KL}), \eta_g = \sqrt{KS}$, and $\rho = \mathcal{O}(\frac{1}{\sqrt{R}})$, iterates satisfy:

$$\mathcal{O}\left( \frac{FL}{\sqrt{RKS}} + \frac{\sqrt{K}G^2}{\sqrt{RS}} + \frac{L^2\sigma_l^2}{R^{3/2}K} + \frac{\sqrt{K}L}{B\sqrt{RS}} + \frac{\sqrt{K}\det(\boldsymbol{H})}{B\sqrt{RS}L'} \right)$$

By analyzing the convergence terms of FedSAM, we observed $\det(\boldsymbol{H})$ and $B$ explicitly emerge in the rate. These results demonstrate that the convergence rate of FedSAM is directly accessed by $\det(\boldsymbol{H})$ and $B$. This aligns with our theoretical bounds, minimal $\det(\boldsymbol{H})$ and maximal $B$ jointly diminish $I(X;G)$), while concurrently stabilizing optimization. Thus, FedSAM achieves stable convergence and enhances robustness against GIA, underscoring the role of curvature reduction.

In summary, the convergence analysis confirms that $\det(\boldsymbol{H})$ and $B$, identified from Theorem 3.1 and Theorem 3.2, contribute both robustness and optimization. The proof is given in Appendix B.3.

## 4 EXPERIMENTS

### 4.1 EXPERIMENTAL SETUPS

**Setup:** We have conducted main experiments on the CIFAR-10 (Krizhevsky et al., 2009) classification task. Across all experiments, we have used a randomly initialized LeNet (Zhu et al., 2019), a simple baseline architecture for FL. The FL scenario consists of one server and 10 clients. In each round, 5 clients are randomly selected to participate in each round. We run 1,000 rounds in total. We consider Independent and Identically Distributed (IID) cases of data partitioning across clients. The hyperparameters are provided in the Appendix C. All reconstruction attacks targeted the same image drawn from the training set, and the target gradient used for reconstruction was computed with a batch size of $B = 1$, using a single minibatch over a single epoch. For all attacks, we first inferred labels using the procedure proposed by iDLG (Zhao et al., 2020) and subsequently used them during the attack. We calculated four different metrics —PSNR(Peak Signal to Noise Ratio), SSIM(Structural Similarity Index Map), LPIPS(Learned Perceptual Image Patch Similarity), and MSE(Mean Squared Error) —to evaluate the difference between the reconstructed image and the ground-truth image. A pre-trained VGG network is used to calculate the LPIPS score.

**Gradient inversion attack baselines:** We evaluated vulnerability to gradient inversion using optimization-based attacks **iDLG** (Zhao et al., 2020), **GI** (Yin et al., 2021), **IG** (Geiping et al., 2020) and generative-model attacks dubbed **GIAS** (Jeon et al., 2021) and **GIFD** (Fang et al., 2023). For these generative model–based attacks, we used a pretrained StyleGAN2 (Karras et al., 2020) to generate high-quality images with sufficient fidelity.

**SAM baselines:** We have adopted various FL plus SAM methods and compared them with FedAvg. We use **FedSAM** (Qu et al., 2022a), **FedASAM** (Caldarola et al., 2022), **FedGF** (Lee & Yoon, 2024) for baselines. We additionally test **Sharpness Aware Initialization (SAI)** (Wang et al., 2025) in FL, which provides a flat initialization.

### 4.2 RELATIONSHIP BETWEEN HESSIAN AND GRADIENT INVERSION ATTACK

For a baseline FedAvg, we estimate the Hessian and perform gradient inversion attacks at every 100 rounds to investigate the relationship between the Hessian values and the reconstruction quality. The empirical Hessian matrix was computed as the average of the Hessian matrix over 100 samples drawn from the training set. Since Hessian $\mathbf{H}$ can be indefinite, directly evaluating $\log \det(\mathbf{H})$ may be numerically unstable. We therefore adopt a sign-invariant and positive semi-definite argument surrogate: $\log \det(\mathbf{H}+\lambda I) \approx \frac{1}{2} \mathrm{tr}\left[ \log\left(\mathbf{H}^T H + \lambda I\right)\right]$, where damping $\lambda > 0$ ensures $\mathbf{H}^2 + \lambda I \succ 0$. In this experiment, we fixed damping $\lambda = 10^{-3}$.

**Analysis on the relationship between Hessian and GIA:** Figure 1a shows the PSNR score of the reconstructed image by GIA methods over training rounds, while Figure 1b shows the $\log$ determinant value of the empirical Hessian of global models at the same rounds. PSNR begins at the peaks around 0 with a randomly scratched model and reduces as the training round proceeds, eventually plateauing with only minor variation. The $\log$ determinant value of Hessian follows a similar tendency. It has the highest value at the initial state and stabilizes after decreasing. Moreover, this

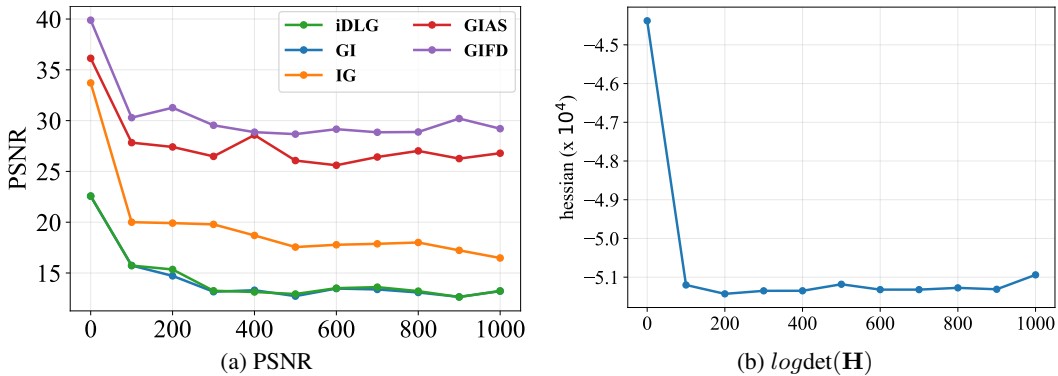

Figure 1: (a) PSNR scores between reconstructed image and ground truth. (b) $\log$ determinant of Hessian over rounds

tendency aligns with previous observations that reconstruction quality (measured by PSNR) drops in trained networks. The similar tendency between Hessian and the reconstruction quality is empirical evidence of **Theorem** 3.1, which states that the mutual information between input data and the gradient is upper bounded by the $\log$ determinant value of the empirical Hessian term. The results for other reconstruction metrics are shown in Appendix D.

### 4.3 FL with Flat Minima Searching improves robustness against GIAs

We demonstrate the reconstruction fidelity by performing GIAs on the models trained with SAM-based FL methods. As previous results indicate that early rounds are more susceptible to GIAs, we focus particularly on rounds from 1 to 200, which are considered risky. All attacks are repeated every 10-round interval within rounds 1 to 200, averaging over 10 target samples. For a fair comparison, all attacks target the same local client and the same target samples.

**SAM settings:** For FedSAM, FedGF, and FedSAI[1], we set the perturbation radius to $\rho = 0.2$, while $\rho = 0.3$ for FedASAM. The step size $\eta = 0.9$ equals to the learning rate of FedAvg. Fixed interpolation coefficient $c = 0.1$ is used for FedGF. All other FL configurations follow the details described in Section 4.1. For FedSAI, we employ 10,000 images from CIFAR-100 as an auxiliary dataset to find a flat initialization. This initialization step runs for 5 epochs with a batch size of 64. We then replace and reinitialize the fully connected layer so that its output dimensionality aligns with that of CIFAR-10, and perform FedSAM to preserve the initial flatness.

**Comparison with SAM-based FL methods:** As shown in Table 1 and Figure 2, flat minima are shown to effectively suppress the GIA's reconstruction across all metrics and rounds. For instance, in round 100, the IG in FedSAM drops PSNR (lower is preferred) from 23.98 to 13.17 ($-45.1\%$ reduction) and increases LPIPS (higher is preferred) from 0.160 to 0.502 ($\times 3.14$ improvement). GIFD demonstrates PSNR 31.46 and LPIPS 0.039 for FedAvg, but it becomes worse for FedAvg, such that PSNR 21.65 ($-31.2\%$) and LPIPS 0.295 ($\times 7.56$). In round 200, PSNR of IG decrease from 21.62 to 12.89 ($-40.4\%$) and LPIPS rises from 0.238 to 0.512 ($\times 2.15$). In case of GIFD, PSNR reduced from 29.65 to 22.74 ($-23.3\%$), while LPIPS increases from 0.082 to 0.226 ($\times 2.76$).

These empirical outcomes are conjectured by the fact that SAM suppresses large eigenvalues and sharp directions, thereby reducing the determinant of the Hessian. According to **Theorem** 3.1, a smaller Hessian determinant indicates that less information is encoded within the gradients. The consistent robustness shown in Table 1 and Figure 2 is clear evidence that the suppression of $I(G; X)$ via flat minima searching is a principled way to resist against various types of GIAs.

Consistent with this interpretation, FedSAM and FedGF manifest comparable characteristics, while FedASAM exhibits marginally stronger robustness than both algorithms. Although FedSAI is initialized from a flatter region via an auxiliary dataset, dataset mismatch sustains a large gradient norm in the initial round, resulting in overall robustness that remains comparable to FedSAM.

---

[1]We utilizes a flat inialization by SAI Wang et al. (2025) as the random scratch of FL, so-called FedSAI.

Table 1: Reconstruction metrics over communication rounds for federated learning methods

| Defense / Attack | | Round 0 | | | | | Round 100 | | | | | Round 200 | | | | |
|---|---|---|---|---|---|---|---|---|---|---|---|---|---|---|---|---|
| | | iDLG | GI | IG | GIAS | GIFD | iDLG | GI | IG | GIAS | GIFD | iDLG | GI | IG | GIAS | GIFD |
| FedAvg | PSNR ↓ | 22.58 | 22.57 | 33.71 | 36.25 | 39.80 | 15.31 | 15.29 | 23.98 | 28.63 | 31.46 | 14.61 | 14.55 | 21.62 | 28.85 | 29.65 |
| | LPIPS ↑ | 0.203 | 0.204 | 0.032 | 0.014 | 0.006 | 0.426 | 0.428 | 0.160 | 0.085 | 0.039 | 0.462 | 0.463 | 0.238 | 0.081 | 0.082 |
| | SSIM ↓ | 0.770 | 0.769 | 0.955 | 0.979 | 0.989 | 0.446 | 0.443 | 0.791 | 0.898 | 0.948 | 0.385 | 0.382 | 0.671 | 0.906 | 0.906 |
| | MSE[†] ↑ | 0.696 | 0.699 | 0.0710 | 0.0320 | 0.0140 | 3.28 | 3.34 | 0.609 | 0.235 | 0.109 | 4.01 | 4.08 | 1.57 | 0.267 | 0.313 |
| FedSAM | PSNR ↓ | 16.16 | 16.17 | 16.55 | 26.12 | 25.79 | 13.47 | 13.47 | 13.17 | 22.77 | 21.65 | 12.92 | 12.97 | 12.89 | 23.94 | 22.74 |
| | LPIPS ↑ | 0.428 | 0.428 | 0.419 | 0.164 | 0.175 | 0.492 | 0.492 | 0.502 | 0.220 | 0.295 | 0.516 | 0.514 | 0.512 | 0.183 | 0.226 |
| | SSIM ↓ | 0.504 | 0.505 | 0.521 | 0.828 | 0.821 | 0.391 | 0.391 | 0.378 | 0.733 | 0.709 | 0.332 | 0.333 | 0.334 | 0.768 | 0.735 |
| | MSE[†] ↑ | 2.49 | 2.49 | 2.27 | 0.262 | 0.292 | 4.63 | 4.63 | 4.90 | 0.627 | 0.702 | 5.15 | 5.09 | 5.19 | 0.429 | 0.551 |
| FedASAM | PSNR ↓ | 14.08 | 14.09 | 16.05 | 22.99 | 22.32 | 11.65 | 11.85 | 12.57 | 20.40 | 17.96 | 12.44 | 12.58 | 13.94 | 21.32 | 20.12 |
| | LPIPS ↑ | 0.515 | 0.515 | 0.459 | 0.244 | 0.270 | 0.559 | 0.556 | 0.535 | 0.275 | 0.394 | 0.538 | 0.534 | 0.499 | 0.237 | 0.324 |
| | SSIM ↓ | 0.391 | 0.391 | 0.477 | 0.734 | 0.723 | 0.288 | 0.295 | 0.330 | 0.671 | 0.570 | 0.309 | 0.313 | 0.370 | 0.658 | 0.603 |
| | MSE[†] ↑ | 3.98 | 3.97 | 2.57 | 0.556 | 0.645 | 7.04 | 6.73 | 5.76 | 0.983 | 1.78 | 5.80 | 5.61 | 4.08 | 0.834 | 1.05 |
| FedGF | PSNR ↓ | 16.18 | 16.18 | 19.33 | 26.05 | 25.77 | 13.64 | 13.63 | 14.66 | 22.11 | 21.26 | 13.03 | 13.02 | 14.45 | 24.12 | 23.11 |
| | LPIPS ↑ | 0.428 | 0.428 | 0.331 | 0.161 | 0.173 | 0.488 | 0.489 | 0.454 | 0.228 | 0.306 | 0.516 | 0.513 | 0.473 | 0.174 | 0.218 |
| | SSIM ↓ | 0.505 | 0.505 | 0.634 | 0.827 | 0.820 | 0.398 | 0.398 | 0.440 | 0.719 | 0.696 | 0.335 | 0.335 | 0.398 | 0.768 | 0.746 |
| | MSE[†] ↑ | 2.48 | 2.48 | 1.23 | 0.267 | 0.290 | 4.45 | 4.47 | 3.50 | 0.725 | 0.762 | 5.04 | 5.04 | 3.73 | 0.434 | 0.506 |
| FedSAI | PSNR ↓ | 16.17 | 16.17 | 16.56 | 26.10 | 25.75 | 13.63 | 13.66 | 13.63 | 22.43 | 20.04 | 14.10 | 14.12 | 13.58 | 23.49 | 21.97 |
| | LPIPS ↑ | 0.428 | 0.428 | 0.419 | 0.161 | 0.177 | 0.482 | 0.481 | 0.477 | 0.213 | 0.333 | 0.493 | 0.491 | 0.493 | 0.216 | 0.268 |
| | SSIM ↓ | 0.505 | 0.505 | 0.521 | 0.830 | 0.819 | 0.391 | 0.392 | 0.399 | 0.734 | 0.653 | 0.391 | 0.392 | 0.372 | 0.766 | 0.694 |
| | MSE[†] ↑ | 2.48 | 2.48 | 2.27 | 0.264 | 0.291 | 4.51 | 4.49 | 4.48 | 0.601 | 1.13 | 4.07 | 4.04 | 4.53 | 0.476 | 0.785 |

[†] MSE value is scaled by $\times 10^2$. ↓: a lower is preferred. ↑: a higher is preferred. Here, 'preferred' means a successful defense.

# 5 ANALYSIS

## 5.1 ANALYSIS ACROSS COMMUNICATION ROUND

Here, we examine the effect of communication rounds on the reconstruction quality of various gradient inversion attacks (GIAs). In particular, we focus on PSNR and related metrics to assess how SAM-based methods compare with FedAvg under different attack strategies.

**PSNR Comparison across Communication Round** Figure 2 illustrates the PSNR trends of each attack mechanism across communication rounds. Consistent with our theoretical framework, which posits that approaches directly reducing Hessian values such as SAM-based methods can enhance defense against various attack mechanisms, Figure 2 shows that SAM-based approaches substantially outperform FedAvg. Even as the number of communication rounds increases, SAM-based methods continue to yield lower PSNR scores, indicating stronger resistance to these attacks.

For example, SAM-based approaches on IG, GIAS, and GIFD methods consistently maintain sufficient gap between relative to FedAvg. The reason is that since mechanisms of IG, GIAS, and GIFD are related to gradient-direction alignment, SAM-based approaches directly suppresses sharp directions, which counterattack the attack mechanisms Similarly, SAM-based approaches outperforms FedAvg against iDLG and GI, that compute Euclidean distance to match gradients. While the performance gap, measured in PSNR, is marginal during training progresses, the consistent advantage of SAM-based methods holds, where it still supports our theoretical expectations.

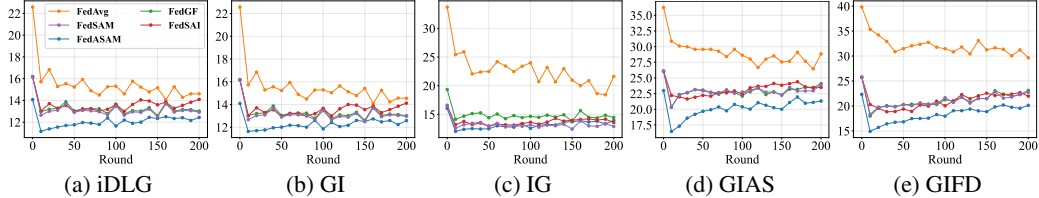

Figure 2: PSNR scores of the reconstructed image by GIAs during rounds

**Convergence Analysis of the Hessians:** Figure 3 illustrates the trajectory of the $\log$-determinant value of empirical Hessian values over 1,000 training rounds for each method. Consistent with previous trends, SAM-based approaches yield lower determinant values compared to FedAvg and exhibit substantially reduced volatility in later rounds. Notably, since FedSAI begins from a smoother loss landscape, it exhibits a lower initial Hessian determinant, which is consistent with our theoretical framework. These results demonstrate that with lower determinant of the Hessian term across rounds. For readability, all reported values have been scaled by $10^{-4}$.

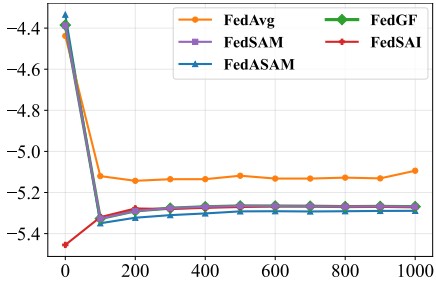

Figure 3: $\log \det(\mathbf{H})$ over rounds.

In Table 1, some attacks achieve a higher reconstruction score in round 200 than round 100. This pattern is consistent with Figure 3. Under SAM, Hessian is significantly low at initialization, rises modestly during an early transient, and then stabilizes. Reconstruction metrics can mirror this trajectory with a brief increase before plateauing, which explains corner cases for rounds around 100.

## 5.2 ABLATION STUDY

We perform ablation experiments to explore the effect of the perturbation radius. We vary the FedSAM perturbation radius $\rho = \{0.05, 0.1, 0.15, 0.2\}$ while keeping all other FL settings, local models, and optimization settings. We here used IG as a representative attack method.

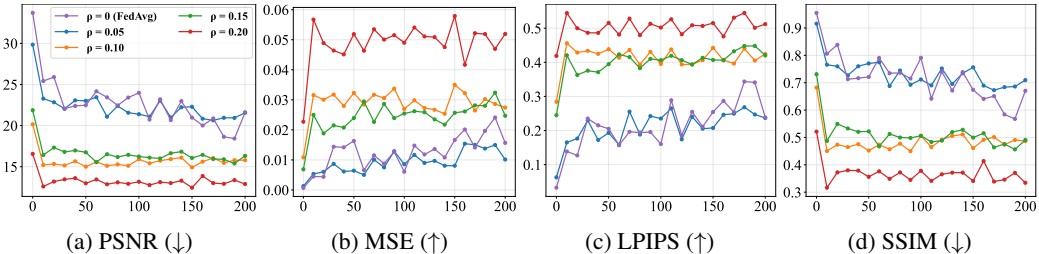

(a) PSNR ($\downarrow$)  (b) MSE ($\uparrow$)  (c) LPIPS ($\uparrow$)  (d) SSIM ($\downarrow$)

Figure 4: **Effect of $\rho$ on each Metrics** This figure illustrates how varying the perturbation radius affects the results. Since $\rho = 0$ is equivalent to the FedAvg, the findings indicate stronger defensive performance as the radius increases, where SAM-based usually outperforms.

**Effect of Perturbation Radius on Reconstruction:** Figure 4 shows the quality of reconstruction with evaluation metrics under varying perturbed radius in FedSAM. As the perturbation radius $\rho$ increases, MSE and LPIPS increase accordingly, while PSNR and SSIM decrease, revealing the degraded attack fidelity. At $\rho = 0.05$, FedSAM produces reconstruction quality akin to FedAvg across metrics and indicates that flattening curvature is minimal.

With increases in $\rho$, the results consistently reveal a gradual degradation in data reconstruction across all evaluation metrics. To further understand this phenomenon, we measure Hessian under varying values $\rho$, as illustrated in Figure 5, which captures the loss landscape. In SAM, a larger perturbation radius $\rho$ broadens the parameter exploration space, thereby encouraging the model to avoid high-curvature directions and consequently enhancing the likelihood of convergence toward flatter minima. As a result of this tendency, the gradient encodes less information about the input data, thereby leading to the mitigation of GIA.

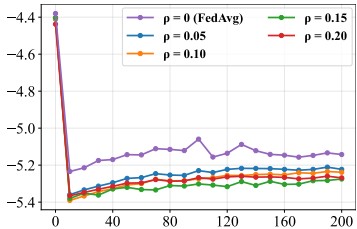

Figure 5: Effect of $\rho$ on Hessian

**Effect of batch size:** Note that the above analysis considers a single batch. However, in a practical FL setting, clients typically train their model with multiple batches. To evaluate the effect of batch size, we conducted experiments on CIFAR-10 with batch size $B = \{1, 2, 4\}$ under both FedAvg and FedSAM at round 0, where GIA occurs frequently. As shown in Figure 6, enlarging the batch size reduces reconstruction fidelity, demonstrating that a larger batch size mitigates the success of such attacks. Moreover, compare FedSAM to FedAvg, applying SAM with expanded batch size alleviates GIA, supporting the insight in Theorem 3.2.

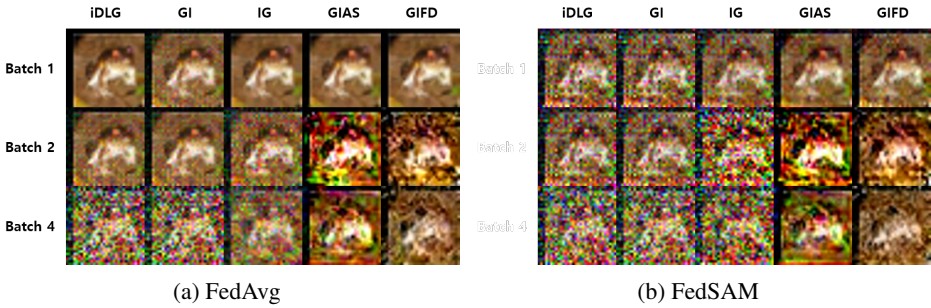

Figure 6: Effect of batch size on the performance of GIA

## 5.3 QUALITATIVE RESULTS

In Figure 7, we visually present the qualitative results of the reconstructed image samples. Among the traditional optimization-based attacks, we select GI Yin et al. (2021) and IG Geiping et al. (2020), where GI shows the minimal gap between FedAvg and SAM-based approaches, while IG demonstrates the most significant gap. We observe a clear advantage of SAM-based approaches beyond FedAvg, where their gap is slightly amplified in IG cases. For the recent generative model-based attacks, we investigate GIAS Jeon et al. (2021) and GIFD Fang et al. (2023). We found that FedAvg and SAM-based approaches are both shown to be slightly vulnerable to these attacks, but we confirm the qualitative gains of SAM-based approaches over FedAvg. Notably, FedASAM marginally outperforms FedSAM in the GIAS case.

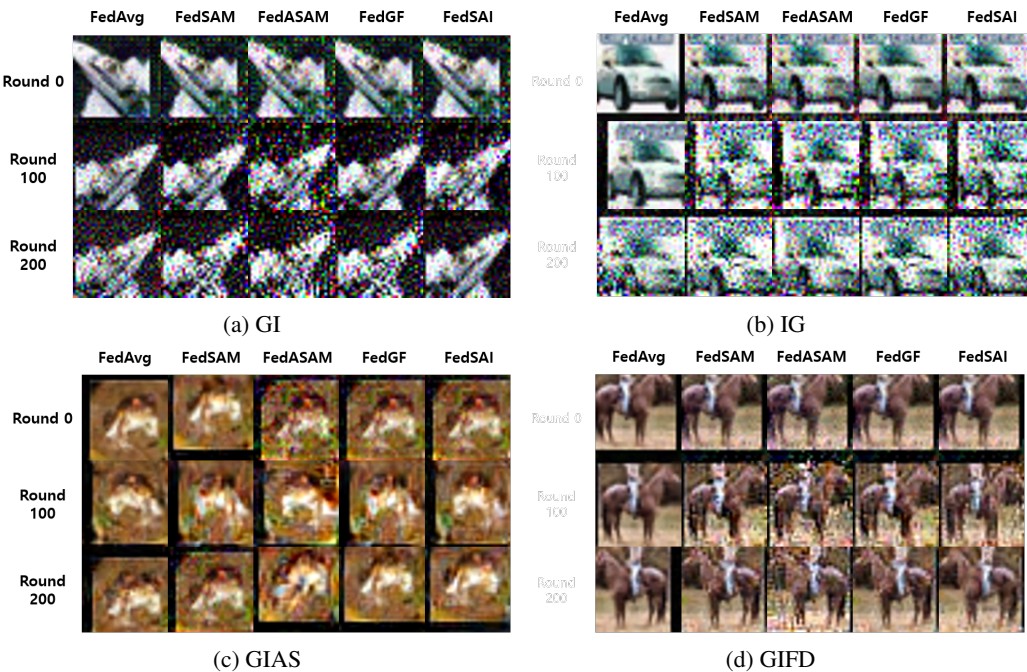

Figure 7: Visualization of images reconstructed by SAM-based federated learning approaches

## 6 CONCLUSION

In this work, we investigate robustness against Gradient Inversion Attacks (GIAs) from the perspective of searching for flat minima. To provide a principled understanding, we present both theoretical and empirical results, specifically, highlighting a flatter minima searching-based FL algorithm that tightens the information encoded in gradients and hinders GIA. Furthermore, our analysis indicates that convergence toward flatter minima simultaneously suppresses mutual information, thereby mitigating privacy leakage, and guarantees stable optimization of model. This demonstrates that searching for flat minima is an effective strategy in FL, by enhancing both model utility and robustness.

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

## A DISCUSSION ON THE ROBUSTNESS OF GRADIENT INVERSION ATTACKS

Recent Gradient Inversion Attack (GIA) mechanisms have substantially increased the difficulty of maintaining model privacy. In response, a parallel line of research has focused on strengthening defense strategies, with an emphasis on the notion of robustness against GIA. In this work, we revisit the robustness against GIA from a new perspective grounded in the geometry of the loss landscape. Specifically, we develop a theoretical framework centered on loss-surface flatness and we apply Sharpness-Aware Minimization (SAM) as one of the most representative approaches. While SAM was originally introduced to improve generalization by smoothing the loss landscape, we are, to our best knowledge, the first to explicitly connect SAM to the study of the robustness against GIA.

Our empirical findings show that SAM yields measurable gains in robustness against GIA, suggesting that flatness is indeed correlated with counterattack of GIA. However, because SAM is not the only mechanism through which flatness can be achieved, current SAM-based approaches have inherent limitations when treating flat-minima search as a sufficient defense strategy, particularly against advanced generative model–based GIAs. Therefore, a key direction for future research is to move beyond gradient- or data-level manipulations and instead develop principled, loss-curvature–based defense mechanisms that can explicitly counter GIA attacks.

## B MATHEMATICAL PROOFS

### B.1 PROOF OF BOUND ON MUTUAL INFORMATION WITH HESSIAN AND BATCH SIZE

We consider the empirical risk function

$$F(\theta) := \sum_{i}^{N} \frac{m_i}{m} F_i(\theta), \qquad F_i(\theta) := \mathbb{E}_{\xi \sim D_i}\big[\mathcal{L}(\theta; \xi)\big],$$

and the aggregated gradient is given by $G = \nabla_\theta F(\theta)$. Since $G$ is deterministically computed from the training batch $X$, the conditional entropy vanishes, and thus the mutual information reduced to $I(X; G) = H(G)$. Among all distributions with a given covariance, the multivariate Gaussian distribution maximizes the differential entropy. Therefore, the entropy of $G$ is upper-bounded by that of Gaussian distribution with mean $\mu$ and covariance $\Sigma_G$, i.e., $I(X; G) \leq H(\mathcal{N}(\mu, \Sigma_G))$.

To express the covariance in terms of the score function, we define the corresponding per-sample score function as $s_i = \nabla_\theta \log p(y \mid x; \theta)$. Then, the covariance of stochastic gradient estimation with batch size $B$ can be expressed as: $\Sigma_G = \frac{1}{B^2} \sum_{i=1}^{B} \text{Cov}(-s_i) = \frac{1}{B} \mathbb{E}[ss^\top]$. To handle the connection between the Fisher information and the Hessian of the loss function, we differentiate the score function as follows: $\mathbb{E}[\nabla_\theta s] = \mathbb{E}\left[\frac{\nabla_\theta^2 p(y|x;\theta)}{p(y|x;\theta)}\right] - \mathbb{E}\left[\left(\frac{\nabla_\theta p(y|x;\theta)}{p(y|x;\theta)}\right)^2\right] = -\mathbb{E}\left[\left(\frac{\nabla_\theta p(y|x;\theta)}{p(y|x;\theta)}\right)^2\right] = -\mathbb{E}[ss^T]$, where the expectation of the score function vanishes by the Leibniz integral rule.

Since the per-sample Hessian is defined as $\boldsymbol{H}(\theta; \xi) = \nabla_\theta^2 \mathcal{L}(\theta; \xi)$, we obain $\mathbb{E}[ss^\top] = \mathbb{E}[\boldsymbol{H}(\theta; \xi)]$. Finally, recalling the empirical Hessian, $\boldsymbol{H} = \sum_{i}^{N} \frac{m_i}{m} \nabla_\theta^2 \mathcal{L}(\theta, \xi_i)$, we conclude that the gradient covariance scales with the empirical Hessian

$$\Sigma_G \approx \frac{1}{B} \boldsymbol{H}.$$

From the differential entropy of a multivariate Gaussian and the correlation of covariance and Hessian, we obtain the bound stated in Theorem 3.1.

### B.2 PROOF OF MUTUAL INFORMATION BOUND VIA SHARPNESS-AWARE MINIMIZATION

Recall that $F_{\text{SAM}}(\theta) := \max_{\|\delta\|_2 \leq \rho} F(\theta + \delta)$, which denotes empirical risk under SAM objective. Since FedSAM seeks the worst-case perturbation with $\|\delta\| \leq \rho$, we set $\delta = \rho v$ with $\|v\| = 1$ to represent the maximization direction. By the second-order Taylor expansion of $F(\theta + \rho v)$, we obtain

$$F(\theta + \rho v) = F(\theta) + \rho G^\top v + \frac{\rho^2}{2} v^\top \boldsymbol{H} v + \mathcal{O}(\rho^3),$$

Here, $\mathcal{O}(\rho^3)$ is the higher-order remainder term arising from the Lipschitz continuity of the Hessian and we omit this higher-order term.

To capture the worst-case scenario, we consider the direction aligned with the largest curvature of F, that is, the unit eigenvector $v_{\max}$ corresponding to the largest eigenvalue $\lambda_{\max}(\boldsymbol{H})$. This leads to the following

$$F_{\text{SAM}}(\theta) \geq F(\theta) + \rho G^\top v_{\max} + \frac{\rho^2}{2}\lambda_{\max}(\boldsymbol{H})$$

However, the alignment between $G$ and $v_{\max}$ is generally anonymous. To preserve generality, we bound the inner product by $G^\top u_{max} \geq -\|G\|_2$ to provide a tight lower bound in the worst case. Substituting this bound to the Taylor expansion inequality, we acquire

$$\lambda_{max}(\boldsymbol{H}) \leq \frac{2}{\rho^2}\Big(F_{SAM}(\theta) - F(\theta) + \rho\|G\|_2\Big).$$

Finally, let $\{\lambda_i\}_{i=1}^d$ be the eigenvalue of $\boldsymbol{H}$. The upper bound of $I(X;G)$ is formulated as follows

$$\frac{d}{2}\log\left(\frac{2\pi e}{B}\right) + \frac{1}{2}\log\big(\det(\boldsymbol{H})\big) \overset{(a)}{=} \frac{d}{2}\log\left(\frac{2\pi e}{B}\right) + \frac{1}{2}\log\Big(\prod_{i=1}^d \lambda_i\Big)$$

$$\overset{(b)}{=} \frac{d}{2}\log\left(\frac{2\pi e}{B}\right) + \frac{d}{2}\log\left(\Big(\prod_{i=1}^d \lambda_i\Big)^{\frac{1}{d}}\right)$$

$$\overset{(c)}{\leq} \frac{d}{2}\log\left(\frac{2\pi e}{B}\right) + \frac{d}{2}\log\left(\frac{1}{d}\sum_{i=1}^d \lambda_i\right)$$

$$\overset{(d)}{\leq} \frac{d}{2}\log\left(\frac{2\pi e}{B}\right) + \frac{d}{2}\log\big(\lambda_{\max}(\boldsymbol{H})\big).$$

where (c) proceeds from the AM-GM inequality and (d) handles the fact that the maximum eigenvalue upper bounds the average. This completes the proof of Theorem 3.2.

### B.3    Proof of Convergence Analysis

#### B.3.1    Preliminary Assumptions, Lemmas and Description of FedSAM

We recall the following lemmas and assumptions from (Qu et al., 2022b) and omit their proofs, as these are provided in detail in (Qu et al., 2022b).

**Lemma 1** *(Relaxed triangle inequality) Let $\{v_1, \ldots, v_\tau\}$ be $\tau$ vectors in $\mathbb{R}^d$. Then, the following are true: (1) $\|v_i + v_j\|^2 \leq (1+\alpha)\|v_i\|^2 + \big(1 + \frac{1}{\alpha}\big)\|v_j\|^2$ for any $\alpha > 0$, and (2) $\left\|\sum_{i=1}^\tau v_i\right\|^2 \leq \tau\sum_{i=1}^\tau \|v_i\|^2$.*

**Lemma 2** *For random variables $x_1, \ldots, x_n$, we have*

$$\mathbb{E}\big[\|x_1 + \cdots + x_n\|^2\big] \leq n\,\mathbb{E}\big[\|x_1\|^2 + \cdots + \|x_n\|^2\big].$$

**Lemma 3** *For independent, mean $0$ random variables $x_1, \ldots, x_n$, we have*

$$\mathbb{E}\big[\|x_1 + \cdots + x_n\|^2\big] = \mathbb{E}\big[\|x_1\|^2 + \cdots + \|x_n\|^2\big].$$

**Lemma 4** *(Separating mean and variance for SAM) The stochastic gradient $\nabla F_i(\theta, \xi_i)$ computed by the $i$-th client at model parameter $\theta$ using minibatch $\xi$ is an unbiased estimator of $\nabla F_i(\theta)$ with variance bounded by $\sigma_l^2$. The gradient of SAM is formulated by*

$$\mathbb{E}\left[\Big\|\sum_{k=0}^{K-1} g_{i,k}^r\Big\|^2\right] \leq K\sum_{k=0}^{K-1}\mathbb{E}\big[\|\nabla F_i(\theta_{i,k}^r)\|^2\big] + \frac{KL^2\rho^2}{N}\sigma_l^2$$

$$\mathbb{E}\left[\Big\|\sum_{k=0}^{K-1} g_{i,k}^r\Big\|^2\right] \leq K\sum_{k=0}^{K-1}\mathbb{E}\big[\|\nabla F_i(\theta_{i,k}^r)\|^2\big] + KL^2\rho^2\sigma_l^2.$$

From the shared global parameters $\theta^{r-1}$, the local updates for $k \in [K]$ are given by

$$\tilde{\theta}_{i,k}^r = \theta_{i,k-1}^r + \rho \frac{g_{i,k-1}^r}{\|g_{i,k-1}^r\|} \qquad \theta_{i,k}^r = \theta_{i,k-1}^r - \eta_l \tilde{g}_{i,k-1}^r.$$

After $K$ times local epochs, where the update $\Delta_i^r = \theta_{i,K}^r - \theta^r$ procured, the server aggregates update from clients $i \in S^r$ with global learning rate $\eta_g$,

$$\Delta^{r+1} = \frac{1}{S} \sum_{i \in \mathcal{S}^r} \Delta_i^r, \qquad \theta^{r+1} = \theta^r + \eta_g \Delta^r.$$

**Assumption 1** *(Smoothness) Each local empirical risk $F_i$ satisfy L-smooth, i.e.,*

$$\|\nabla F_i(\theta) - \nabla F_i(\theta')\| \leq L\|\theta - \theta'\|.$$

*for all $\theta, \theta'$ in its domain and $i \in [N]$.*

**Assumption 2** *(Bounded variance of global gradient without perturbation) Without perturbation $\delta_i$, the global variability of the local gradient of the loss function is bounded by $\sigma_g^2$, i.e.,*

$$\|\nabla F_i(\theta^r) - \nabla F(\theta^r)\|^2 \leq \sigma_g^2, \quad \forall i \in [N] \text{ and } \forall r.$$

**Assumption 3** *(Bounded variance of stochastic gradient) The stochastic gradient $\nabla F_i(\theta, \xi_i)$, computed by the $i$-th client of model parameter $\theta$ using mini-batch $\xi_i$ of size B, is an unbiased estimator of $\nabla F_i(\theta)$ with variance bounded by $\sigma_l^2$, i.e.,*

$$\mathbb{E}_{\xi_i} \left\| \frac{\nabla F_i(\theta, \xi_i)}{\|\nabla F_i(\theta, \xi_i)\|} - \frac{\nabla F_i(\theta)}{\|\nabla F_i(\theta)\|} \right\|^2 \leq \sigma_l^2, \quad \forall i \in [N].$$

*where the expectation is over all local datasets.*

**Lemma 5** *(Bounded $\mathcal{E}_\delta$ of FedSAM) Suppose Assumptions 1-2 hold. Then, for any $\eta_l \leq \frac{1}{4KL}$, drift due to $\delta_{i,k} - \delta$ satisfies*

$$\mathcal{E}_\delta = \frac{1}{N} \sum_i \mathbb{E}\big[\|\delta_{i,k} - \delta\|^2\big] \leq 2K^2 \beta^2 \eta_l^2 \rho^2.$$

**Lemma 6** *(Bounded $\mathcal{E}_\theta$ of FedSAM) Suppose Assumptions 1-2 hold. Then, for any $\eta_l \leq \frac{1}{10KL}$, the drift due to $\theta_{i,k} - \theta$ satisfies*

$$\mathcal{E}_\theta = \frac{1}{N} \sum_i \mathbb{E}\big[\|\theta_{i,k} - \theta\|^2\big] \leq 5K\eta_l^2\Big(2L^2\rho^2\sigma_l^2 + 6K(3\sigma_g^2 + 6L^2\rho^2)\Big) + 6K\|\nabla f(\tilde{\theta})\|^2 + 24K^3\eta_l^4 L^4 \rho^2.$$

### B.3.2 CONVERGENCE ANALYSIS FEDSAM WITH FULL CLIENT PARTICIPANT

To present the convergence of full client participant, we adopt lemmas from (Qu et al., 2022b) and modify them under the empirical risk minimization formulation.

**Lemma 7**

$$\big\langle \nabla F(\tilde{\theta}^r), \mathbb{E}_r\big[\Delta^r + \eta_l K \nabla F(\tilde{\theta}^r)\big]\big\rangle \leq \frac{\eta_l K}{2}\|\nabla F(\tilde{\theta}^r)\|^2 + K\eta_l L^2 \mathcal{E}_\theta + K\eta_l L^2 \mathcal{E}_\delta - \frac{\eta_l}{2KN^2}\mathbb{E}_r\Big\|\sum_{i,k} \nabla F_i(\tilde{\theta}_{i,k})\Big\|^2.$$

**Lemma 8** *For the full client participation scheme, we can bound $\mathbb{E}[\|\Delta^r\|^2]$ as follows:*

$$\mathbb{E}_r\big[\|\Delta^r\|^2\big] \leq \frac{K\eta_l^2 L^2 \rho^2}{N}\sigma_l^2 + \frac{\eta_l^2}{N^2}\left[\Big\|\sum_{i,k} \nabla F_i(\tilde{\theta}_{i,k}^r)\Big\|^2\right] + \frac{\eta_l^2 K}{B}\Big((d-1)K + \frac{\det(\boldsymbol{H})}{L'} + dL_H\rho\Big).$$

*Proof.* For the full client participation scheme, we have:

$$\mathbb{E}_r[\|\Delta^r\|^2] \overset{(a)}{\leq} \frac{\eta_l^2}{N^2}\mathbb{E}_r\left[\left\|\sum_{i,k}\tilde{g}_{i,k}^r\right\|^2\right] \overset{(b)}{=} \frac{\eta_l^2}{N^2}\mathbb{E}_r\left[\left\|\sum_{i,k}\left(\tilde{g}_{i,k}^r - \nabla F_i(\tilde{\theta}_{i,k}^r)\right)\right\|^2\right] + \frac{\eta_l^2}{N^2}\mathbb{E}_r\left[\left\|\sum_{i,k}\nabla F_i(\tilde{\theta}_{i,k}^r)\right\|^2\right]$$

$$\overset{(c)}{\leq} \frac{K\eta_l^2 L^2\rho^2}{N}\sigma_l^2 + \frac{\eta_l^2}{N^2}\left\|\mathbb{E}_r\left[\sum_{i,k}\nabla F_i(\tilde{\theta}_{i,k}^r)\right]\right\|^2 + \frac{\eta_l^2}{N^2}\mathrm{Tr}\left(\mathrm{Cov}\left(\sum_{i,k}\nabla F_i(\tilde{\theta}_{i,k}^r)\right)\right)$$

$$\overset{(d)}{\leq} \frac{K\eta_l^2 L^2\rho^2}{N}\sigma_l^2 + \frac{\eta_l^2}{N^2}\left\|\mathbb{E}_r\left[\sum_{i,k}\nabla F_i(\tilde{\theta}_{i,k}^r)\right]\right\|^2 + \frac{\eta_l^2}{N^2}\mathrm{Tr}\left(\mathrm{Cov}\left(\sum_{i,k}\tilde{g}_{i,k}\right)\right)$$

$$\overset{(e)}{=} \frac{K\eta_l^2 L^2\rho^2}{N}\sigma_l^2 + \frac{\eta_l^2}{N^2}\left\|\mathbb{E}_r\left[\sum_{i,k}\nabla F_i(\tilde{\theta}_{i,k}^r)\right]\right\|^2 + \eta_l^2 K\,\mathrm{Tr}(\tilde{\Sigma}_G)$$

$$\overset{(f)}{\leq} \frac{K\eta_l^2 L^2\rho^2}{N}\sigma_l^2 + \frac{\eta_l^2}{N^2}\left\|\mathbb{E}_r\left[\sum_{i,k}\nabla F_i(\tilde{\theta}_{i,k}^r)\right]\right\|^2 + \frac{\eta_l^2}{B}K\,\mathrm{Tr}\left(\boldsymbol{H}(\tilde{\theta})\right)$$

$$\overset{(g)}{\leq} \frac{K\eta_l^2 L^2\rho^2}{N}\sigma_l^2 + \frac{\eta_l^2}{N^2}\left\|\mathbb{E}_r\left[\sum_{i,k}\nabla F_i(\tilde{\theta}_{i,k}^r)\right]\right\|^2 + \frac{\eta_l^2}{B}K\,\mathrm{Tr}\left(\boldsymbol{H}(\theta) + dL_H\rho\right)$$

$$\overset{(h)}{\leq} \frac{K\eta_l^2 L^2\rho^2}{N}\sigma_l^2 + \frac{\eta_l^2}{N^2}\left\|\mathbb{E}_r\left[\sum_{i,k}\nabla F_i(\tilde{\theta}_{i,k}^r)\right]\right\|^2 + \frac{\eta_l^2 K}{B}\left((d-1)L + \frac{\det(\boldsymbol{H})}{L'} + dL_H\rho\right).$$

Here, (a) and (b) are based on Lemma 2 and Lemma 3, respectively, (c) follows from Lemma 4 and the bias-variance decomposition $\mathbb{E}[\|Z\|^2] = \|\mathbb{E}[Z]\|^2 + \mathrm{Tr}(\mathrm{Cov}(Z))$, (d) employs that $\tilde{g}_{i,k}$ is an unbiased estimator of $\nabla F_i(\tilde{\theta}_{i,k})$.

For step (e), by assuming $\sum_{i,k}\tilde{g}_{i,k} = N\sum_k\tilde{g}_k$ with $\mathrm{Cov}(\tilde{g}_k) = \tilde{\Sigma}_G$, we obtain $\mathrm{Tr}\left(\mathrm{Cov}\left(\sum_{i,k}\tilde{g}_{i,k}\right)\right) = N^2\,\mathrm{Tr}\left(\mathrm{Cov}\left(\sum_k\tilde{g}_k\right)\right) = N^2 K\,\mathrm{Tr}(\tilde{\Sigma}_G)$. Therefore, the third term becomes $\eta_l^2 K\,\mathrm{Tr}(\tilde{\Sigma}_G)$.

Step (f) follows $\Sigma_G \approx \frac{1}{B}\boldsymbol{H}$ in Theorem 3.1 which associates the covariance to empirical Hessian. And step (g) applies $L_H$ Lipschitz continuity of the Hessian in Theorem 3.1.

Let $\boldsymbol{H} \succeq 0$ with eigenvalues $\lambda_d \geq \cdots \geq \lambda_1 \geq 0$. Assume L-smoothness (Assumption 1), where $L$ denote as the maximum Lipschitz constant, i.e., $\boldsymbol{H} \preceq LI$. Then each eigenvalue is bounded by $\lambda_i \preceq L$, thus the sum of the smallest $d-1$ eigenvalues satisfies $\sum_{i=1}^{d-1}\lambda_i \leq \sum_{i=1}^{d-1}L = (d-1)L$. Moreover, we define $L' := \prod_{i=1}^{d-1}\lambda_i(\boldsymbol{H})$. Since $\det(\boldsymbol{H}) = \left(\prod_{i=1}^{d-1}\lambda_i\right)\lambda_d = L'\lambda_d$, we obtain $\mathrm{Tr}(\boldsymbol{H}) = \sum_{i=1}^{d-1}\lambda_i + \lambda_d \leq (d-1)L + \lambda_d = (d-1)L + \frac{\det(\boldsymbol{H})}{L'}$. By substituting the bound $\mathrm{Tr}(\boldsymbol{H}) \leq (d-1)L + \frac{\det(\boldsymbol{H})}{L'}$ into step (g) yields (h).

**Lemma 9** *(Descent Lemma) For all $r \in R-1$ and $i \in S^r$, the iterates generated by FedSAM satisfy:*

$$\mathbb{E}_r\left[F(\tilde{\theta}^{r+1})\right] \leq F(\tilde{\theta}^r) - K\eta_g\eta_l\left(\frac{1}{2} - 30K^2L^2\eta_l^2\right)\left\|\nabla F(\tilde{\theta}^r)\right\|^2$$

$$+ K\eta_g\eta_l\left(10KL^4\eta_l^2\rho^2\sigma_l^2 + 90K^2L^2\eta_l^2\sigma_g^2 + 180K^2L^4\eta_l^2\rho^2 + 120K^4L^6\eta_l^6\rho^2\right.$$

$$\left. + 16K^3\eta_l^4 L^6\rho^2 + \frac{\eta_g\eta_l L^3\rho^2}{N}\sigma_l^2\right) + \frac{(d-1)}{2BK} + \frac{\det(\boldsymbol{H})}{2BKL^d} + \frac{dL_H\rho}{2BKL}.$$

*Proof*

$$\mathbb{E}_r\Big[F(\tilde\theta^{\,r+1})\Big] \le F(\tilde\theta^{\,r}) + \mathbb{E}_r\Big\langle \nabla F(\tilde\theta^{\,r}), \tilde\theta^{\,r+1} - \tilde\theta^{\,r}\Big\rangle + \frac{L}{2}\mathbb{E}_r\Big\|\tilde\theta^{\,r+1} - \tilde\theta^{\,r}\Big\|^2$$

$$\overset{(a)}{=} F(\tilde\theta^{\,r}) + \mathbb{E}_r\Big\langle \nabla F(\tilde\theta^{\,r})\,,\, -\Delta^r + K\eta_g\eta_l\nabla F(\tilde\theta^{\,r}) - K\eta_g\eta_l\nabla F(\tilde\theta^{\,r})\Big\rangle + \frac{L}{2}\eta_g^2\,\mathbb{E}_r[\|\Delta^r\|^2]$$

$$\overset{(b)}{=} F(\tilde\theta^{\,r}) - K\eta_g\eta_l\big\|\nabla F(\tilde\theta^{\,r})\big\|^2 + \eta_g\big\langle \nabla F(\tilde\theta^{\,r}), \mathbb{E}_r[-\Delta^r + K\eta_l\nabla F(\tilde\theta^{\,r})]\big\rangle + \frac{L}{2}\eta_g^2\,\mathbb{E}_r[\|\Delta^r\|^2]$$

$$\overset{(c)}{\le} F(\tilde\theta^{\,r}) - \frac{K\eta_g\eta_l}{2}\big\|\nabla F(\tilde\theta^{\,r})\big\|^2 + K\eta_g\eta_l L^2\mathcal{E}_\theta + K\eta_g\eta_l L^2\mathcal{E}_\delta + \frac{\eta_g\eta_l}{2KN}\mathbb{E}_r\Big[\Big\|\sum_{i,k}\nabla F_i(\tilde\theta^{\,r}_{i,k})\Big\|^2\Big]$$

$$\qquad + \frac{L}{2}\eta_g^2\,\mathbb{E}_r\big[\|\Delta^r\|^2\big]$$

$$\overset{(d)}{\le} F(\tilde\theta^{\,r}) - \frac{K\eta_g\eta_l}{2}\big\|\nabla F(\tilde\theta^{\,r})\big\|^2 + K\eta_g\eta_l L^2\mathcal{E}_\theta + K\eta_g\eta_l L^2\mathcal{E}_\delta + \frac{K\eta_g^2\eta_l^3 L^3\rho^2}{N}\sigma_l^2$$

$$\qquad + \frac{KL}{2B}\eta_l^2\eta_g^2\Big((d-1) + \frac{\det(\boldsymbol{H})}{L'} + dL_H\rho\Big)$$

$$\overset{(e)}{\le} F(\tilde\theta^{\,r}) - K\eta_g\eta_l\Big(\frac{1}{2} - 30K^2L^2\eta_l^2\Big)\big\|\nabla F(\tilde\theta^{\,r})\big\|^2$$

$$\qquad + K\eta_g\eta_l\Big(10KL^4\eta_l^2\rho^2\sigma_l^2 + 90K^2L^2\eta_l^2\sigma_g^2 + 180K^2L^4\eta_l^2\rho^2 + 120K^4L^6\eta_l^6\rho^2 + 16K^3\eta_l^4L^6\rho^2 + \frac{\eta_g\eta_l L^3\rho^2}{N}\sigma_l^2\Big)$$

$$\qquad + \frac{(d-1)}{2BK} + \frac{\det(\boldsymbol{H})}{2BKL^d} + \frac{dL_H\rho}{2BKL}.$$

Here, (a) is from the iterate update of FedSAM, (b) is required from the unbiased estimators, (c) relies on Lemma 7, (d) follows Lemma 8, and (e) holds under the learning rate $\eta_l\eta_g \le \frac{1}{KL}$.

By applying the telescoping sum of the result in Lemma 9 for $r = [R]$ with learning rate conditions $\eta_l = \frac{1}{\sqrt{R}KL}, \eta_g = \sqrt{KN}$ and perturbation amplitude $\rho = \frac{1}{\sqrt{R}}$ yields

$$\frac{1}{R}\sum_{r=1}^{R}\mathbb{E}\|F(\theta^{r+1})\|$$

$$= \mathcal{O}\Big( \frac{FL}{\sqrt{RKN}} + \frac{\sigma_g^2}{R} + \frac{L^2\sigma_l^2}{R^{3/2}\sqrt{KN}} + \frac{L^2}{R^{3/2}} + \frac{L}{BK^{3/2}\sqrt{RN}} + \frac{\det(\boldsymbol{H})}{BK^{3/2}L'\sqrt{RN}} + \frac{L_H}{BK^{3/2}R\sqrt{N}} \Big)$$

Note that both $\frac{L^2}{R^{3/2}}$ term and $\frac{L_H}{BK^{3/2}R\sqrt{N}}$ term decay faster with $R$ than $\frac{L}{BK^{3/2}\sqrt{RN}}$ and $\frac{\det(\boldsymbol{H})}{BK^{3/2}L'\sqrt{RN}}$. Hence, in large $R$, these terms are asymptotically negligible. In addition, the term $\frac{L_H}{BK^{3/2}R\sqrt{N}}$ appears due to the Lipschitz continuity of Hessian and contributes only marginally to robustness and convergence. By dropping faster decaying terms in $R$, we can obtain the following.

$$\frac{1}{R}\sum_{r=1}^{R}\mathbb{E}\|F(\theta^{r+1})\| = \mathcal{O}\Big( \frac{FL}{\sqrt{RKN}} + \frac{\sigma_g^2}{R} + \frac{L^2\sigma_l^2}{R^{3/2}\sqrt{KN}} + \frac{L}{BK^{3/2}\sqrt{RN}} + \frac{\det(\boldsymbol{H})}{BK^{3/2}L'\sqrt{RN}} \Big)$$

*Proof.* Multiplying $\frac{1}{CK\eta_l\eta_g R}$ on both sides with $\left(\frac{1}{2} - 30K^2L^2\eta_l^2\right) > C > 0$ if $\eta_l < \frac{1}{\sqrt{30}KL}$, we obtain

$$\frac{1}{R}\sum_{r=1}^{R}\mathbb{E}\left[\left\|\nabla F(\theta^{r+1})\right\|^2\right]$$

$$\leq \frac{F(\tilde{\theta}^r) - F(\tilde{\theta}^{r+1})}{CK\eta_g\eta_l R}$$

$$+ \frac{1}{C}\left(10KL^4\eta_l^2\rho^2\sigma_l^2 + 90K^2L^2\eta_l^2\sigma_g^2 + 180K^2L^4\eta_l^2\rho^2 + 120K^4L^6\eta_l^6\rho^2 + 16K^3\eta_l^4L^6\rho^2 + \frac{\eta_g\eta_l L^3\rho^2}{N}\sigma_l^2\right)$$

$$+ \frac{1}{CK\eta_g\eta_l R}\left(\frac{(d-1)}{2BK} + \frac{\det(\boldsymbol{H})}{2BKL'} + \frac{dL_H\rho}{2BKL}\right)$$

$$\leq \frac{F(\tilde{\theta}^0) - F^\star}{CK\eta_g\eta_l R}$$

$$+ \frac{1}{C}\left(10KL^4\eta_l^2\rho^2\sigma_l^2 + 90K^2L^2\eta_l^2\sigma_g^2 + 180K^2L^4\eta_l^2\rho^2 + 120K^4L^6\eta_l^6\rho^2 + 16K^3\eta_l^4L^6\rho^2 + \frac{\eta_g\eta_l L^3\rho^2}{N}\sigma_l^2\right)$$

$$+ \frac{1}{CK\eta_g\eta_l R}\left(\frac{(d-1)}{2BK} + \frac{\det(\boldsymbol{H})}{2BKL'} + \frac{dL_H\rho}{2BKL}\right)$$

$$= \frac{F}{CK\eta_g\eta_l R}$$

$$+ \frac{1}{C}\left(10KL^4\eta_l^2\rho^2\sigma_l^2 + 90K^2L^2\eta_l^2\sigma_g^2 + 180K^2L^4\eta_l^2\rho^2 + 120K^4L^6\eta_l^6\rho^2 + 16K^3\eta_l^4L^6\rho^2 + \frac{\eta_g\eta_l L^3\rho^2}{N}\sigma_l^2\right)$$

$$+ \frac{1}{CK\eta_g\eta_l R}\left(\frac{(d-1)}{2BK} + \frac{\det(\boldsymbol{H})}{2BKL'} + \frac{dL_H\rho}{2BKL}\right)$$

where $F$ equals to $F(\tilde{\theta}^0) - F^\star$.

### B.3.3 CONVERGENCE ANALYSIS FEDSAM WITH PARTIAL CLIENT PARTICIPANT

To establish the convergence properties under the partial client participant, we adapt lemmas introduced in (Qu et al., 2022b) and adjust to our empirical risk and Hessian framework.

**Lemma 10** *For the partial client participation with $S \subseteq N$, the variance is bounded as $\mathbb{E}_r[\|\Delta^r\|^2]$:*

$$\mathbb{E}_r\left[\|\Delta^r\|^2\right] \leq \frac{K\eta_l^2 L^2\rho^2}{S}\sigma_l^2 + \frac{S}{N}\sum_i\left\|\sum_{j=1}^{K-1}\nabla F_i(\tilde{\theta}_{i,k}^r)\right\|^2 + \frac{S(S-1)}{N^2}\left\|\sum_{j=0}^{K-1}\nabla F_i(\tilde{\theta}_{i,j}^r)\right\|^2.$$

**Lemma 11** *Suppose $\nabla F_i(\tilde{\theta}_{i,k})$ for all $k \in [K]$ and $i \in [N]$ is chosen according to FedSAM, we have*

$$\sum_i\mathbb{E}\left[\left\|\sum_k\nabla F_i(\tilde{\theta}_{i,k})\right\|^2\right] \leq 30NK^2L^2\eta_l^2\left(2L^2\rho^2\sigma_l^2 + 6K(3\sigma_g^2 + 6L^2\rho^2) + 6K\|\nabla F(\tilde{\theta})\|^2\right)$$

$$+ 144K^4L^6\eta_l^4\rho^2 + 12NK^4L^2\eta_l^2\rho^2 + 3NK^2(3\sigma_g^2 + 6L^2\rho^2) + 3NK^2\|\nabla F(\tilde{\theta})\|^2.$$

Let local and global learning rates $\eta_l$ and $\eta_g$ be $\eta_l \leq \frac{1}{10KL}$, $\eta_l\eta_g \leq \frac{1}{KL}$. From descent lemma, the convergence under partial client participant is obtained as follows.

*Proof.*

$$\mathbb{E}_r\left[F(\tilde{\theta}^{r+1})\right]$$

$$\overset{(a)}{\leq} F(\tilde{\theta}^r) - \frac{K\eta_g\eta_l}{2}\left\|\nabla F(\tilde{\theta}^r)\right\|^2 + K\eta_g\eta_l L^2\mathcal{E}_\theta + K\eta_g\eta_l L^2\mathcal{E}_\delta - \frac{\eta_g\eta_l}{2KN}\mathbb{E}_r\left[\left\|\sum_{i,k}\nabla F_i(\tilde{\theta}^r_{i,k})\right\|^2\right] + \frac{L}{2}\eta_g^2\mathbb{E}_r\left[\|\Delta^r\|^2\right]$$

$$\overset{(b)}{\leq} F(\tilde{\theta}^r) - \frac{K\eta_g\eta_l}{2}\left\|\nabla F(\tilde{\theta}^r)\right\|^2 + K\eta_g\eta_l L^2\mathcal{E}_\theta + K\eta_g\eta_l L^2\mathcal{E}_\delta + \frac{K\eta_g^2\eta_l^3 L^3\rho^2}{S}\sigma_l^2$$

$$- \frac{\eta_g\eta_l}{2KN}\mathbb{E}_r\left[\left\|\sum_{i,k}\nabla F_i(\tilde{\theta}^r_{i,k})\right\|^2\right] + \frac{\eta_g^2 LS}{2N}\sum_i\left\|\sum_{j=1}^{K-1}\nabla F_i(\tilde{\theta}_{i,k})\right\|^2 + \frac{\eta_g^2 LS(S-1)}{2N^2}\left\|\sum_{j=0}^{K-1}\nabla F_i(\tilde{\theta}_{i,j})\right\|^2$$

$$\overset{(c)}{\leq} F(\tilde{\theta}^r) - \frac{K\eta_g\eta_l}{2}\left\|\nabla F(\tilde{\theta}^r)\right\|^2 + K\eta_g\eta_l L^2\mathcal{E}_\theta + K\eta_g\eta_l L^2\mathcal{E}_\delta + \frac{K\eta_g^2\eta_l^3 L^3\rho^2}{2S}\sigma_l^2 + \frac{L\eta_g^2\eta_l^2}{2NS}\sum_i\|\sum_k\nabla F_i(\tilde{\theta}^r_{i,k})\|^2$$

$$\overset{(d)}{\leq} F(\tilde{\theta}^r) - \frac{K\eta_g\eta_l}{2}\left\|\nabla F(\tilde{\theta}^r)\right\|^2 + K\eta_g\eta_l L^2\mathcal{E}_\theta + K\eta_g\eta_l L^2\mathcal{E}_\delta + \frac{K\eta_g^2\eta_l^3 L^3\rho^2}{N}\sigma_l^2 + \frac{L\eta_g^2\eta_l^2}{2NS}K\sum_{i,k}\|\nabla F_i(\tilde{\theta}^r_{i,k})\|^2$$

$$\overset{(e)}{\leq} F(\tilde{\theta}^r) - \frac{K\eta_g\eta_l}{2}\left\|\nabla F(\tilde{\theta}^r)\right\|^2 + K\eta_g\eta_l L^2\mathcal{E}_\theta + K\eta_g\eta_l L^2\mathcal{E}_\delta + \frac{K\eta_g^2\eta_l^3 L^3\rho^2}{N}\sigma_l^2$$

$$+ \frac{L\eta_g^2\eta_l^2}{2S}K\sum_k\left[\left\|\frac{1}{N}\sum_{i=1}^N\nabla F_i(\tilde{\theta}^r_{i,k})\right\|^2 + \text{Tr}\left(\text{Cov}\left(\nabla F_i(\tilde{\theta}^r_{i,k})\right)\right)\right]$$

$$\overset{(f)}{\leq} F(\tilde{\theta}^r) - \frac{K\eta_g\eta_l}{2}\left\|\nabla F(\tilde{\theta}^r)\right\|^2 + K\eta_g\eta_l L^2\mathcal{E}_\theta + K\eta_g\eta_l L^2\mathcal{E}_\delta + \frac{K\eta_g^2\eta_l^3 L^3\rho^2}{N}\sigma_l^2$$

$$+ \frac{L^2\eta_g^2\eta_l^2}{2S}\frac{K}{N}\sum_i\mathbb{E}\left[\left\|\sum_k\nabla F_i(\tilde{\theta}_{i,k})\right\|^2\right] + \frac{L\eta_g^2\eta_l^2 K^2}{2BS}\left((d-1)L + \frac{\det(\boldsymbol{H})}{L'} + dL_H\rho\right)$$

$$\overset{(g)}{\leq} F(\tilde{\theta}^r) - K\eta_g\eta_l\left(\frac{1}{2} - 30K^2L^2\eta_l^2 - \frac{L\eta_g\eta_l}{2S}\left(3K + 180K^3L^2\eta_l^2\right)\right)\left\|\nabla F(\tilde{\theta}^r)\right\|^2$$

$$+ K\eta_g\eta_l\left(10KL^4\eta_l^2\rho^2\sigma_l^2 + 90K^2L^2\eta_l^2\sigma_g^2 + 180K^2L^4\eta_l^2\rho^2 + 120K^4L^6\eta_l^6\rho^2 + 16K^3\eta_l^4L^6\rho^2 + \frac{L^3\eta_g\eta_l\rho^2}{2S}\sigma_l^2\right)$$

$$+ \frac{K^2\eta_g^2\eta_l^2}{S}\left(30KL^5\eta_l^2\rho^2\sigma_l^2 + 180K^2L^3\eta_l^2\rho^2 + 360KL^5\eta_l^2\rho^2 + 72K^3L^7\eta_l^4\rho^2 + 6K^3L^3\eta_l^2\rho^2 + 6KL\sigma_g^2 + 6KL^3\rho^2\right)$$

$$+ \frac{K^2L\eta_g^2\eta_l^2}{2BS}\left((d-1)L + \frac{\det(\boldsymbol{H})}{L'} + dL_H\rho\right)$$

Here, (a) is based on Lemma 9, (b) from Lemma 10, (c) considering the expectation of $r$-th that $KL\eta_g\eta_l \leq \frac{S-1}{S}$, (d) applies Lemma 2, (e) is based on bias-variance decomposition $\mathbb{E}[\|Z\|^2] = \|\mathbb{E}[Z]\|^2 + \text{Tr}(\text{Cov}(Z))$, (f) applies the step (h) in Lemma 8, (g) combines Lemmas 5, 6 and 11.

For partial client participation, considering the telescope sum over $R$ communication rounds with local and global learning rates $\eta_l = \frac{1}{\sqrt{R}KL}$, $\eta_g = \sqrt{KS}$ and perturbation $\rho = \frac{1}{\sqrt{R}}$, we obtain

$$\frac{1}{R}\sum_{r=1}^R\mathbb{E}\|F(\theta^{r+1})\|$$

$$= \mathcal{O}\left(\frac{FL}{\sqrt{R}KS} + \frac{\sqrt{K}G^2}{\sqrt{R}S} + \frac{L^2\sigma_l^2}{R^{3/2}K} + \frac{\sqrt{K}L^2}{R^{3/2}\sqrt{S}} + \frac{\sqrt{K}L}{B\sqrt{R}S} + \frac{\sqrt{K}\det(\boldsymbol{H})}{B\sqrt{R}SL'} + \frac{\sqrt{K}L_H}{BR\sqrt{S}}\right)$$

*Proof.* Multiplying $\frac{1}{CK\eta_l\eta_g R}$ on both sides, $\left(\frac{1}{2} - 30K^2L^2\eta_l^2 - \frac{L\eta_g\eta_l}{2S}(3K + 180K^3L^2\eta_l^2)\right) > C > 0$, on the descent lemma of partial client, we obtain

$$\frac{1}{R}\sum_{r=1}^{R}\mathbb{E}\Big[\big\|F(\tilde{\theta}^{r+1})\big\|\Big] \le \frac{F(\tilde{\theta}^{r}) - F(\tilde{\theta}^{r+1})}{CK\eta_g\eta_\ell R}$$

$$+ \frac{1}{C}\Big(10KL^4\eta_\ell^2\rho^2\sigma_\ell^2 + 90K^2L^2\eta_\ell^2\sigma_g^2 + 180K^2L^4\eta_\ell^2\rho^2 + 120K^4L^6\eta_\ell^6\rho^2 + 16K^3\eta_\ell^4L^6\rho^2 + \frac{L^3\eta_g\eta_\ell\rho^2}{2S}\sigma_\ell^2\Big)$$

$$+ \frac{\eta_g\eta_\ell}{S}\Big(30KL^5\eta_\ell^2\rho^2\sigma_\ell^2 + 180K^2L^3\eta_\ell^2\rho^2 + 360KL^5\eta_\ell^2\rho^2 + 72K^3L^7\eta_\ell^4\rho^2 + 6K^3L^3\eta_\ell^2\rho^2 + 6KL\sigma_g^2 + 6KL^3\rho^2\Big)$$

$$+ \frac{KL\eta_g\eta_l}{2BS}\Big((d-1)L + \frac{\det(\boldsymbol{H})}{L'} + dL_H\rho\Big)$$

$$\le \frac{F}{CK\eta_g\eta_\ell R}$$

$$+ \frac{1}{C}\Big(10KL^4\eta_\ell^2\rho^2\sigma_\ell^2 + 90K^2L^2\eta_\ell^2\sigma_g^2 + 180K^2L^4\eta_\ell^2\rho^2 + 120K^4L^6\eta_\ell^6\rho^2 + 16K^3\eta_\ell^4L^6\rho^2 + \frac{L^3\eta_g\eta_\ell\rho^2}{2S}\sigma_\ell^2\Big)$$

$$+ \frac{\eta_g\eta_\ell}{S}\Big(30KL^5\eta_\ell^2\rho^2\sigma_\ell^2 + 180K^2L^3\eta_\ell^2\rho^2 + 360KL^5\eta_\ell^2\rho^2 + 72K^3L^7\eta_\ell^4\rho^2 + 6K^3L^3\eta_\ell^2\rho^2 + 6KL\sigma_g^2 + 6KL^3\rho^2\Big)$$

$$+ \frac{KL\eta_g\eta_l}{2BS}\Big((d-1)L + \frac{\det(\boldsymbol{H})}{L'} + dL_H\rho\Big)$$

where, the second equality employ $F = F(\tilde{\theta}_0) - F^* \le F(\tilde{\theta}_r) - F(\tilde{\theta}_{r+1})$.

Note that the terms $\frac{\sqrt{K}L^2}{R^{3/2}\sqrt{S}}$ and $\frac{\sqrt{K}L_H}{BR\sqrt{S}}$ vanish faster in $R$ than $\frac{\sqrt{K}L}{B\sqrt{RS}}$ and $\frac{\sqrt{K}\det(\boldsymbol{H})}{B\sqrt{RS}L'}$. Moreover, the term $\frac{\sqrt{K}L_H}{BR\sqrt{S}}$ is introduced to account for Hessian $L_H$-Lipschitz continuity and provides a minor contribution to robustness and convergence. Thus, we obtain the simplified rate as follows.

$$\frac{1}{R}\sum_{r=1}^{R}\mathbb{E}\|F(\theta^{r+1})\| = \mathcal{O}\Big(\frac{FL}{\sqrt{RKS}} + \frac{\sqrt{K}G^2}{\sqrt{RS}} + \frac{L^2\sigma_l^2}{R^{3/2}K} + \frac{\sqrt{K}L}{B\sqrt{RS}} + \frac{\sqrt{K}\det(\boldsymbol{H})}{B\sqrt{RS}L'}\Big)$$

## C  EXPERIMENTAL SETTING

### C.1  HYPERPARAMETERS AND FL SETTING

We provide all training hyperparameters and federated learning settings used in our main experiments as below.

Table 2: Hyperparameters and FL settings

| Hyperparameters | Value |
| --- | --- |
| Datasets $D$ | CIFAR-10 |
| Local model $M$ | LeNet |
| Clients $K$ | 10 |
| Clients per round $m$ | 5 |
| Rounds $R$ | 1000 |
| Local epochs $E$ | 5 |
| Batch size $B$ | 64 |
| Learning rate $\eta$ | 0.01 |
| Optimizer | SGD |
| Weight decay $\lambda$ | 0.0005 |
| Non-IID $\alpha$ | 10(IID) |

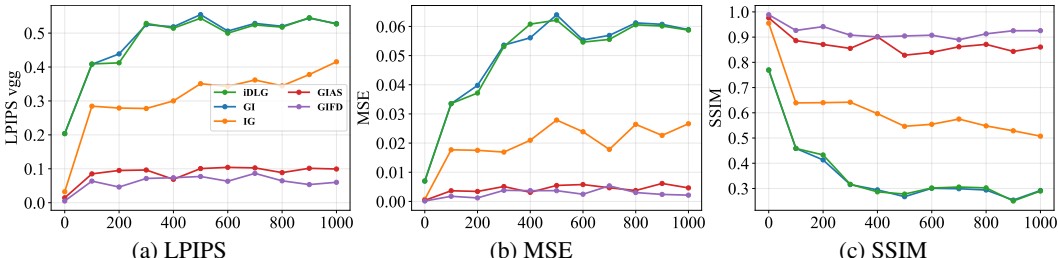

Figure 8: LPIPS, MSE, SSIM scores reconstructed by all of gradient inversion attacks in FedAvg over communication rounds.

# D  ADDITIONAL EXPERIMENTAL RESULTS

## D.1  RECONSTRUCTION METRICS OVER ROUNDS

Figure 8 shows the evolution of gradient inversion attack performance under FedAvg across the communication round, evaluated by LPIPS, MSE, and SSIM. The three metrics exhibit tendencies consistent with PSNR. As communication rounds progress, LPIPS and MSE increase while SSIM decreases, indicating that gradient inversion attacks become more difficult.

## D.2  ADDITIONAL COMPARISON

We evaluated the reconstruction quality of the gradient inversion attacks using four metrics. Section 4.3 provides an analysis using the PSNR score. We utilize the remaining three metrics to analyze the effect of SAM on the gradient inversion attack.

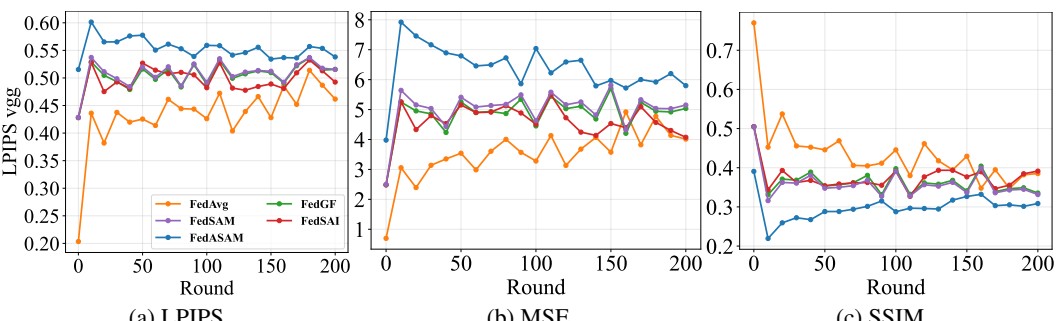

Figure 9: LPIPS, MSE, SSIM scores reconstructed by iDLG in all federated learning methods over communication rounds.

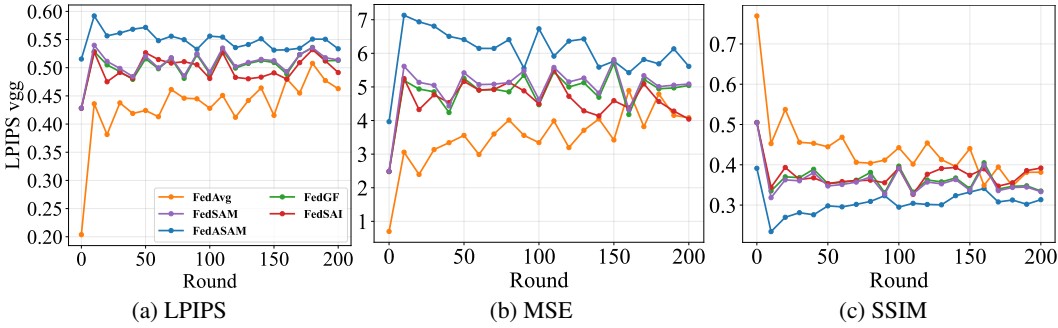

Figure 10: LPIPS, MSE, SSIM scores reconstructed by GI in all federated learning methods over communication rounds.

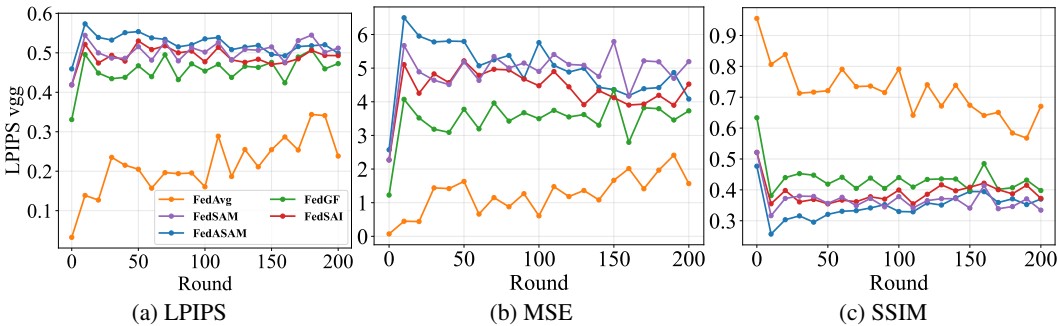

Figure 11: LPIPS, MSE, SSIM scores reconstructed by IG in all federated learning methods over communication rounds.

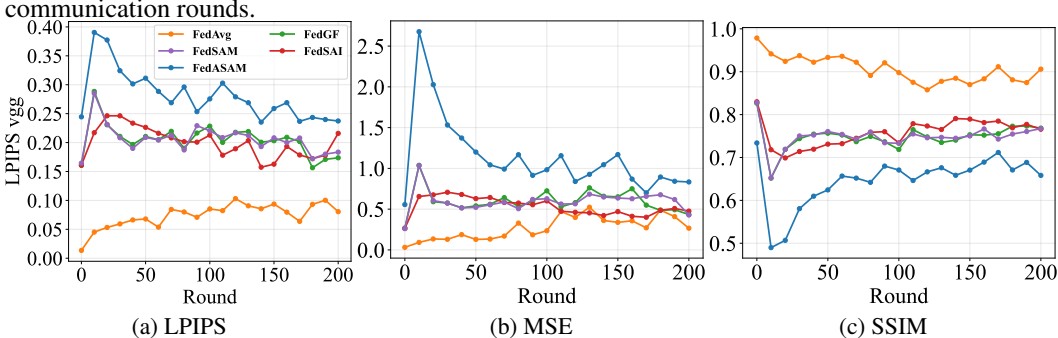

Figure 12: LPIPS, MSE, SSIM scores reconstructed by GIAS in all federated learning methods over communication rounds.

Figure 9 - 13 show that FedAvg achieves lower LPIPS and MSE and higher SSIM compared to SAM-based federated learning methods, demonstrating that SAM makes gradient inversion attack difficult.

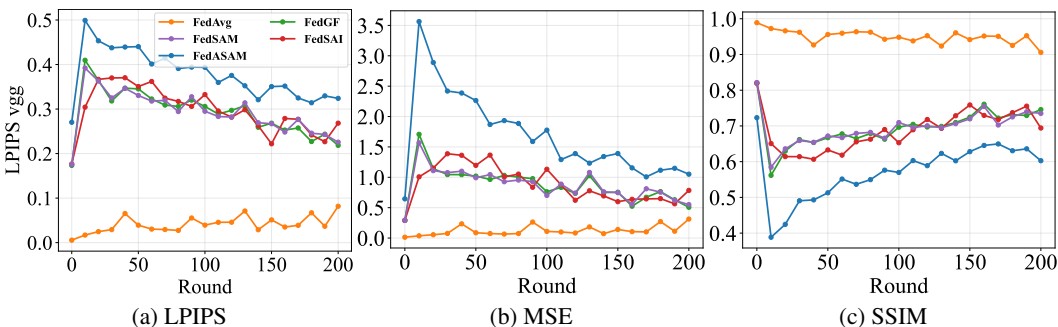

(a) LPIPS  (b) MSE  (c) SSIM

Figure 13: LPIPS, MSE, SSIM scores reconstructed by GIFD in all federated learning methods over communication rounds.

