# OpenReview forum: "Understanding Robustness Against Gradient Inversion Attacks: A Flat Minima Perspective"
_ICLR.cc/2026/Conference — ICLR 2026 Conference Withdrawn Submission_

### Official Review · Reviewer_orDq · 2025-10-30

**Soundness:** 2
**Presentation:** 2
**Contribution:** 2
**Rating:** 4
**Confidence:** 4

**Summary:**

This paper investigates the privacy risks posed by GIAs in federated learning scenarios. The authors argue that most existing defenses rely on heuristic gradient modifications and lack a principled understanding of when gradients are vulnerable. To address this gap, they reinterpret GIAs through the lens of mutual information I(X;G)between gradients and data, proving that it is upper-bounded by the loss Hessian. Building on this theory, they analyze flat-minima-search-based FL algorithms, showing that flatter minima inherently suppress Hessian values, thereby reducing information leakage. Extensive experiments demonstrate significant robustness improvements against GIAs, highlighting flat minima’s dual benefits for both generalization and privacy preservation.

**Strengths:**

1.	The paper provides a principled and interpretable understanding of GIAs by introducing a mutual information perspective and explicitly linking gradient vulnerability to the Hessian of the loss function. This connection provides a solid theoretical foundation for analyzing gradient leakage, moving beyond previous ad-hoc or heuristic defenses.
2.	The authors propose a new viewpoint by linking flat minima searching to privacy protection. They argue that optimization toward flatter minima inherently smooths the loss landscape, thereby suppressing the mutual information between data and gradients. This insight reveals that the pursuit of flat minima not only benefits generalization but also acts as an intrinsic defense mechanism against data leakage.
3.	The paper rephrases the convergence process of flat-minima-search algorithms such as FedSAM and provides formal robustness guarantees against GIAs. This theoretical assurance strengthens the credibility of the proposed defense mechanism and bridges the gap between convergence behavior and privacy robustness in FL.

**Weaknesses:**

1. The authors claim that existing approaches that perturb training data or gradients “lack a principled way of knowing when gradients become vulnerable to GIA.” However, it remains unclear whether the effectiveness of these perturbation-based defenses can be interpreted through the proposed mutual information framework between G and X. For example, in the case of differential privacy (DP), under a fixed privacy budget, would I(G+\delta;X) still remain within the theoretical upper bound derived in this paper? How does I(G+\delta;X) compare to I(G;X) is it theoretically larger or smaller? The authors do not discuss these aspects, which makes the paper’s motivation somewhat insufficient.

2. In the experimental section, the authors do not present the impact of flat-minima-search optimization methods on the global model’s accuracy and convergence speed. Intuitively, constraining the upper bound of I(G;X) may hinder the model’s ability to sufficiently learn informative features of inputs from gradients, potentially leading to slower convergence and reduced accuracy. Moreover, the trade-off among model accuracy, convergence efficiency, and privacy protection is an important issue in this research area. The authors should provide a clear discussion on how the flat-minima optimization influences these factors.

3. The authors’ experimental setup for federated learning is impractical. In realistic FL scenarios, client data are typically non-IID because each user holds private datasets with diverse distributions; evaluating only under IID setting therefore risks overestimating robustness and generality. Moreover, the datasets used in the paper appear limited in scale and diversity. The authors should extend experiments to larger and more diverse benchmarks (for example, CIFAR-100 and CelebA) to show that the proposed Hessian-based insights and flat-minima defenses scale beyond small datasets.

4. The authors should provide complete descriptions for the x-axis and y-axis in the line charts presenting the experimental results, especially for Figs. 1, 3, and 4, ensuring that the format is consistent with Fig. 2. Additionally, the labels on the right side of Figs. 5 and 6 are in white, which should be changed to black for better visibility.

**Questions:**

Please refer to Weaknesses.

---

### Official Review · Reviewer_V9LN · 2025-10-30

**Soundness:** 1
**Presentation:** 1
**Contribution:** 1
**Rating:** 0
**Confidence:** 5

**Summary:**

This paper studies an effective way to mitigate gradient inversion attacks, proposes to use a flat minima searching-based algorithm and analyzes from the perspective of mutual information. The paper provides both theoretical analysis and numerical results.

**Strengths:**

* The research topic is quite interesting and useful in federated learning, which aims at performing privacy-preserved distributed machine learning.

**Weaknesses:**

* The paper is hard to read starting from Section 3. The most obvious weakness is that no algorithm pseudocode is provided, even in the appendix. Without pseudocode, the reviewer cannot evaluate the correctness of the algorithm or the analysis.
* It seems that the paper shares a lot of common ground with, both for analysis and problem formulation, Qu et al. (2022a). Then, the question becomes what differentiates this paper from Qu et al. (2022a), other than that this paper deals with a multi-agent system?
* It is also interesting that the main convergence results in lines 215 and 220 are not stated as a theorem(s). Still, it remains unclear what FedSAM is. It is also a bit strange that the authors choose to keep all the assumptions in the appendix, which render many constants in the theorem itself unstated. For example, Lipschitz constant $L$,
* Due to the above reasons, I believe the current paper in its current form is far from ready for review.

**Questions:**

* Can the authors provide algorithm pseudocode?
* Can the authors unbiasedly state their contributions as compared to Qu et al. (2022a)?
* Can the authors organize their theoretical results with a formal statement?
* Can the authors thoroughly reexamine their paper to make sure the terms are used after they are properly stated?

---

### Official Review · Reviewer_sxsy · 2025-10-31

**Soundness:** 3
**Presentation:** 3
**Contribution:** 3
**Rating:** 4
**Confidence:** 4

**Summary:**

This paper investigated the problem of gradient inversion attacks in distributed learning, aiming to understand under what circumstances the gradient is likely to leak the original data from an information theory perspective. Gradient Inversion Attacks are interpreted with mutual information between the gradients and the data, which is revealed to be upper-bounded by the Hessian of loss and connects to the recently proposed sharpness-aware minimization (SAM) approaches. It reveals from both theoretical and empirical aspects that SAM improves robustness against gradient inversion attacks.

**Strengths:**

1. Interpreting gradient inversion attacks with the mutual information between gradients and the data makes sense, and revealing its connection with SAM is interesting.
2. The arguments are validated from both theoretical and experimental perspectives.

**Weaknesses:**

1. The experiments are conducted mainly on small networks, and the effectiveness on larger models is not verified.
2. The major baseline in this paper is FedAvg, which has been shown to be vulnerable to gradient inversion attacks. Without comparison with existing defense mechanisms, it is hard to tell the effectiveness of the proposed method.
3. The comparison with FedAvg may not necessarily be fair, considering that SAM-based approaches essentially require additional SGD steps, which makes gradient inversion attacks more difficult.
4. The model performance in terms of test accuracy should be evaluated as well.

**Questions:**

1. In Table 1, it can be observed that FedASAM performs slightly better than other SAM-based approaches. Could you please explain?
2. Could the authors consider comparing the SAM-based approaches with existing defense mechanisms?
3. Since additional SGD steps increase the difficulty of gradient inversion, it is not clear if worse reconstruction is due to more local steps (i.e., one gradient descent step and one gradient ascent step for SAM) or the flat minima search principle. Could the authors further clarify?

---

### Official Review · Reviewer_gH8Q · 2025-11-01

**Soundness:** 3
**Presentation:** 3
**Contribution:** 2
**Rating:** 2
**Confidence:** 4

**Summary:**

The paper studies gradient inversion attacks in federated learning via mutual information. The authors link the mutual information to the Hessian matrix, suggesting that a flatter loss landscape can enhance privacy protection. They propose utilizing sharpness-aware optimization algorithms, such as SAM, as a defense mechanism. They conduct some experiments to validate this idea.

**Strengths:**

* The paper is well-organized and easy to follow.

* A flatter loss landscape contributes to privacy preservation is intuitive

* The experimental results generally support the authors' core idea regarding the correlation between flatness and defense against GIAs.

**Weaknesses:**

* Studying data leakage in FL using information theory is not fresh [1]. The theoretical results appear to be direct applications or minor reformulations of existing theoretical results regarding the relationship between the Fisher Information, the Hessian, and information leakage. The technical contribution in this regard is limited.

* This paper is not well-suited for Learning Theory track, as it does not delve deeply into core learning theory aspects. It would be a much better fit for Privacy or Adversarial Learning track.

* The authors have missed many recent and crucial studies [1,2]. Some of the attack baselines used in the experiments are somewhat outdated. The paper also does not include defense methods. A more thorough survey is needed.

* The effectiveness of using sharpness-aware optimization as a defense is questionable. Sharpness-aware algorithms often require a significant number of iterations to settle into a flat minimum. An adversary could launch the attack at early time points in the training process (e.g., during initialization or early transient phases) when the loss surface is still sharp.

* The experiments rely on small models and datasets. Moreover, based on Table 1, the defense performance is insufficient. The PSNR values often remain above 20, which may still allow for recognizable image reconstruction and thus, significant privacy leakage.

[1] Defending Against Data Reconstruction Attacks in Federated Learning: An Information Theory Approach

[2] Boosting Gradient Leakage Attacks: Data Reconstruction in Realistic FL Settings

**Questions:**

See Weaknesses.

---

### Note · Authors · 2026-01-18

I have read and agree with the venue's withdrawal policy on behalf of myself and my co-authors.